# Neutralizing and Enhancing Epitopes of the SARS-CoV-2 Receptor-Binding Domain (RBD) Identified by Nanobodies

**DOI:** 10.3390/v15061252

**Published:** 2023-05-26

**Authors:** Kanasap Kaewchim, Kittirat Glab-ampai, Kodchakorn Mahasongkram, Thanatsaran Saenlom, Watayagorn Thepsawat, Monrat Chulanetra, Kiattawee Choowongkomon, Nitat Sookrung, Wanpen Chaicumpa

**Affiliations:** 1Graduate Program in Immunology, Department of Immunology, Faculty of Medicine Siriraj Hospital, Bangkok 10700, Thailand; kanasap.kaw@alumni.mahidol.ac.th; 2Center of Research Excellence in Therapeutic Proteins and Antibody Engineering, Department of Parasitology, Faculty of Medicine Siriraj Hospital, Bangkok 10700, Thailand; kittirat.gla@mahidol.edu (K.G.-a.); kodchakorn.mah@mahidol.ac.th (K.M.); thanatsaran.sae@mahidol.ac.th (T.S.); watayagorn.the@mahidol.edu (W.T.); monrat.chl@mahidol.ac.th (M.C.); nitat.soo@mahidol.ac.th (N.S.); 3Department of Biochemistry, Faculty of Sciences, Kasetsart University, Bangkok 10900, Thailand; fsciktc@ku.ac.th; 4Biomedical Research Incubator Unit, Department of Research, Faculty of Medicine Siriraj Hospital, Mahidol University, Bangkok 10700, Thailand

**Keywords:** neutralizing epitope, enhancing epitope, nanobody (single-domain antibody/VH/V_H_H), phage display, receptor-binding domain (RBD), SARS-CoV-2

## Abstract

Engineered nanobodies (VHs) to the SARS-CoV-2 receptor-binding domain (RBD) were generated using phage display technology. A recombinant Wuhan RBD served as bait in phage panning to fish out nanobody-displaying phages from a VH/V_H_H phage display library. Sixteen phage-infected *E. coli* clones produced nanobodies with 81.79–98.96% framework similarity to human antibodies; thus, they may be regarded as human nanobodies. Nanobodies of *E. coli* clones 114 and 278 neutralized SARS-CoV-2 infectivity in a dose-dependent manner; nanobodies of clones 103 and 105 enhanced the virus’s infectivity by increasing the cytopathic effect (CPE) in an infected Vero E6 monolayer. These four nanobodies also bound to recombinant Delta and Omicron RBDs and native SARS-CoV-2 spike proteins. The neutralizing VH114 epitope contains the previously reported VYAWN motif (Wuhan RBD residues 350–354). The linear epitope of neutralizing VH278 at Wuhan RBD 319RVQPTESIVRFPNITN334 is novel. In this study, for the first time, we report SARS-CoV-2 RBD-enhancing epitopes, i.e., a linear VH103 epitope at RBD residues 359NCVADVSVLYNSAPFFTFKCYG380, and the VH105 epitope, most likely conformational and formed by residues in three RBD regions that are spatially juxtaposed upon the protein folding. Data obtained in this way are useful for the rational design of subunit SARS-CoV-2 vaccines that should be devoid of enhancing epitopes. VH114 and VH278 should be tested further for clinical use against COVID-19.

## 1. Introduction

SARS-CoV-2, the causative agent of the COVID-19 pandemic, uses a trimeric spike (S) glycoprotein that decorates the virion surface to bind to the human angiotensin-converting enzyme 2 receptor (hACE2) for cell entry and replication therein [1]; thus, the S protein is the target of the virus-neutralizing antibodies. The SARS-CoV-2 S protein is synthesized as a 1273-amino-acid (aa) polyprotein precursor (N-terminal 13 aa is a signal sequence (SS)) on the rough endoplasmic reticulum (RER). Like the SARS-CoV S protein, the SS of the SARS-CoV-2 S protein is removed by cellular signal peptidases in the RER lumen [2,3,4,5]. Glycosylation of the S polyprotein occurs co-translationally (N-glycosylation) and in the Golgi complex (O-glycosylation) [6,7]. In the trans-Golgi network, the S protein is cleaved by proteases at residue 685 of the S1/S2 site (^680^SPRRAR↓SV^687^) into S1 and S2 subunits, which remain associated by non-covalent interaction [4,8,9]. The mature S protein contains two subunits, i.e., the S1 subunit (aa 14–685) and S2 subunit (aa 686–1273). The S1 subunit consists of an N-terminal domain (NTD; aa 14–305) and a C-terminal domain (CTD; aa 306–685). The S1 C-terminal domain contains a receptor-binding domain (RBD; aa 319–541), where the receptor-binding motif (RBM; aa 437–508) is located [10]. The S2 subunit consists of a fusion peptide (FP; aa 788–806), heptad repeat 1 (HR1; aa 912–984), central helix (CH; aa 987–1035), connection domain (CD; aa 1080–1163), heptad repeat 2 (HR2; aa 1163–1213), transmembrane domain (TM; aa 1212–1234), and cytoplasmic tail (CT; aa 1213–to the last aa) [10]. All S2 structures function synchronously in the virus–host membrane fusion process for the virus’s genome release into the cytosol and further replication [11]. At the N-terminal to the FP, there is an S2′ site (KPTKR↓SFI, shared by all members of coronavirus genera) to which, after cleaving by the host furin, the FP is exposed to mediate host–virus membrane fusion [12].

One of the safety concerns regarding COVID-19 immunization and immunotherapy is the disease aggravation by antibodies (either from post-vaccination, convalescing, or therapeutic monoclonal antibodies) via antibody-dependent enhancement (ADE) at the new infection. Antibody-dependent enhancement has been demonstrated for several viral infections—notably, flaviviruses, e.g., Dengue virus [13,14] and Zika virus [15,16]; influenza A virus [17]; respiratory syncytial virus (RSV) [18]; and human and animal coronaviruses, including MERS-CoV, SARS-CoV-2, and others [19,20,21,22,23,24,25]. Different mechanisms of ADE have been recognized, including extrinsic ADE, where the Fc fragments of the virus–antibody complexes bind to Fc receptors on mononuclear phagocytes and enhance entry of the complexes into the cells, consequently causing a high viral load [13,14]; the virus–antibody–complement complexes can also enter the cells via the complement receptor. There is also intrinsic ADE, where the infecting virus causes increased suppression of cytokine signaling (SOCs) molecules that inhibit type 1 (Th1) immune response and type 1 interferon production but activate interleukin-10 biosynthesis, thereby favoring a type 2 (Th2) immune response, which heightens the viral production and burst-out [14]. Finally, there is ADE via enhancement of immune activation, where the virus–antibody complexes mediate inflammation through complement activation (formation of anaphylatoxins, chemotactic factors, and membrane-attack complexes) and recruitment of inflammatory and immune cells, causing tissue immunopathology and cytokine storm [19]. The virus–antibody complexes may obscure the virus from being recognized by the intracellular pathogen recognition receptor (PRRs), facilitating viral replication [26]. The sub-neutralizing/non-neutralizing antibodies may enhance the virus’s entry into permissive cells (enhancing antibodies); in the case of SARS-CoV-2, the enhancing antibodies’ crosslinked epitopes are located at the S1 NTDs of adjacent spike trimers and, in the process, promote an upstanding (open) form of the RBD, which enhances viral entry [20]. The monoclonal antibody binds to the spike protein on the surface of MERS-CoV-like particles (VLPs), causing conformational changes of the protein, which becomes prone to proteolytic activation and enhances the viral entry [21]. It has been also suggested that ADE may explain differences in the severity of COVID-19 in different areas, due to prior exposure to similar antigenic epitopes [22]. Therefore, for safety reasons, vaccines against viruses should not contain the epitopes that induce virus-enhancing antibodies. Likewise, therapeutic antibody preparations should contain only neutralizing antibodies.

The COVID-19 pandemic has turned to endemicity, akin to influenza, and may require boosters/annual vaccinations against the new circulating/epidemic variant(s), as well as therapeutics for the severely ill patients—especially those in high-risk groups (e.g., the elderly and those with underlying conditions, such as obesity, pregnancy, cardiovascular diseases, or other chronic diseases). In this study, our original aim was to produce human-like single-domain antibodies (nanobodies) to the receptor-binding domain (RBD) of SARS-CoV-2 using phage display technology for further use as an immunotherapeutic agent for COVID-19 patients. The nanobodies—which are small, engineered antibodies—usually have deep and high penetration efficacy, high solubility, and higher stability under harsh pH or ionic strength compared to intact four-chain antibody molecules [27,28]. The generated RBD-bound nanobodies derived from phage-transfected *E. coli* clones were tested for their neutralizing activity against authentic SARS-CoV-2. It was found that while some of the RBD-bound nanobodies neutralized the SARS-CoV-2 infectivity, the others enhanced the virus’s infectivity. Thus, neutralizing and enhancing epitopes of the SARS-CoV-2 RBD-specific nanobodies were investigated. This information should be useful not only for safe subunit vaccine design (i.e., vaccine component without enhancing epitopes), but also for proper selection of safe monoclonal antibody preparation for passive immunization/immunotherapy against COVID-19.

## 2. Materials and Methods

### 2.1. Cells, Viruses and Viral Propagation

African green monkey kidney (Vero E6) cells were cultured in Dulbecco’s modified Eagle’s medium (DMEM) (Gibco, Thermo Fisher Scientific, Waltham, MA, USA) supplemented with 10% fetal bovine serum (FBS) (HyClone; GE Healthcare Life Sciences, Marlborough, MA, USA), 100 units/mL penicillin, 100 mg/mL streptomycin, and 2 mM L-glutamine (Gibco, Thermo Fisher Scientific) (complete DMEM).

All experiments involving live SARS-CoV-2 were conducted in the BSL-3 laboratory, Faculty of Medicine Siriraj Hospital, Mahidol University, Bangkok, following the regulations of laboratory biosafety. The SARS-CoV-2 isolates used in this study included the Wuhan wildtype and variants of concern (VOCs), i.e., Delta (B.1.617.2) and Omicron (B1.1.529). They were isolated from infected Thai patients. The viruses were propagated as described previously [29]. Briefly, the Vero E6 cells (~4 × 10^6^ cells) were seeded in T75 culture flasks (Nunc, Thermo Fisher Scientific) and incubated at 37 °C in a 5% CO_2_ atmosphere overnight. The culture fluid in each flask was removed, and individual SARS-CoV-2 isolates in 3 mL of plain DMEM were added to the cell monolayer (MOI 0.01). The flasks were incubated as above for 1 h; then, 15 mL of complete DMEM was added and incubated further until the maximal cytopathic effect (CPE) was observed. The culture fluids were collected and centrifuged, and the viral titers in the supernatants were determined by plaque-forming assay (PFA) [29].

### 2.2. Phage Library Displaying Nanobodies (VHs/V_H_Hs)

A *Camelus dromedarius* nanobody (VH/V_H_H) phage display library was constructed previously in our laboratory using peripheral blood mononuclear cells (PBMCs) of an eight-month-old naïve male *C. dromedarius* [30]. Messenger RNAs from the PBMCs were reverse-transcribed to cDNAs. The genes coding for immunoglobulin (Ig) variable heavy-chain domains (VHs/V_H_Hs/nanobodies) of the camel (*vhs*/*vhhs*) were PCR-amplified using the cDNAs as templates. Fourteen forward and three reverse primers specific to all human Ig VH and JH gene families and subfamilies [30] were used for the PCR amplification of the camel *vhs*/*vhhs*. The human Ig forward primers annealed to the 5′ ends of the complementary exons of the camel *vhs*/*vhhs*, and the human reverse primers bound to the 3′ ends of the complementary camel JH exons (*jhs*). The camel gene products that could be amplified by the human Ig primers were ligated into a pCANTAB 5E phagemid vector, and the ligation mixture was introduced into competent TG1 *Escherichia coli* (a fast-growing derivative strain of JM101 *E. coli* that does not modify or has restrictions on the transformed exogenous DNA). The recombinant phagemid-transformed TG1 *E. coli* bacteria were cultured in Luria–Bertani (LB) broth and co-infected with M13KO7 helper phages. A total of ∼4 × 10^11^ VH/V_H_H-displaying mature phage particles were rescued from the supernatant of the TG1 *E. coli* culture. The diversity of the VH/V_H_H-displaying phages in the library was determined by using restriction fragment length polymorphism (RFLP) analysis of the *vhs*/*vhhs* sequences of the representative *E. coli* clones, and more than 80% of the phages revealed different patterns of the VH/V_H_H coding sequences; thus, the antibody diversity of the library was calculated to be ~3.2 × 10^11^ [30].

### 2.3. Production of Nanobodies to the RBD of SARS-CoV-2

Phage panning was performed to select phage clones displaying nanobodies (VHs/V_H_Hs) that bound to the SARS-CoV-2 RBD from the nanobody (VH/V_H_H) phage display library. A recombinant RBD of the SARS-CoV-2 Wuhan wildtype (Fapon Biotech, Dongguan, China) was added into a well of a 96 well-microplate (Nunc, Thermo Fisher Scientific) (1 µg of RBD in 100 µL of phosphate-buffered saline, pH 7.4 (PBS)); a control antigen—1 µg of bovine serum albumin (BSA) in 100 µL of PBS—was added to another well of the plate, and the plate was kept at 4 °C overnight. The coated wells were washed three times with PBS containing 0.05% Tween 20 (PBS-T), blocked with protein-free blocking solution (Pierce™ Protein-Free Blocking Buffer, Thermo Fisher Scientific) at room temperature (25 ± 2 °C) for 1 h, and washed again with PBS-T. Fifty microliters of the nanobody (VH/V_H_H) phage display library (approximately 5 × 10^10^ phage particles) was added to the well coated with BSA for subtraction of the library. After 1 h at 37 °C, the fluid containing BSA-unbound phages was moved to the well coated with the SARS-CoV-2 RBD. After being kept at 37 °C for 1 h, the fluid containing RBD-unbound phages was discarded; the well was washed thoroughly with PBS containing 0.5% (*v*/*v*) Tween 20 before adding 100 µL of mid-log-phase-grown HB2151 *E. coli* (K12 Δ(*lac-pro*), *ara*, *nal*^r^, *thi*/F’[*proAB*, *lacI^q^*, *lacZ*Δ*M15*; lifescience-market.com), and phage transfection was allowed for 30 min. The phage-infected bacteria were spread on 2× YT agar plates supplemented with 100 µg/mL ampicillin and 2% (*w*/*v*) D-glucose (2× YT-AG), and then incubated at 37 °C overnight. The phage-transformed HB2151 *E. coli* cells that grew on the selective agar plates were screened by PCR-based *vh*/*vhh* gene amplification using pCANTAB 5E phagemid-specific primers: forward (R1) 5′-CCATGATTACGCCAAGCTTT-3′ and reverse (R2) 5′-GCTAGATTTCAAAACAGCAGAAAGG-3′.

The *E. coli* transformants carrying the *vh*/*vhh* phagemid vectors were grown in a medium containing a 1 mM final concentration of isopropyl β-d-1-thiogalactopyranoside (IPTG; Vivantis, Selangor, Malaysia). E-tagged nanobodies (VHs/V_H_Hs) in the HB2151 *E. coli* lysates were determined by Western blot analysis, using rabbit anti-E-tag antibody (ab3397, Abcam, Cambridge, United Kingdom) as a tracer. The E-tagged nanobodies contained in the HB2151 *E. coli* lysates were checked for binding to the recombinant RBD protein of the SARS-CoV-2 Wuhan wildtype (homologous antigen) by indirect enzyme-linked immunosorbent assay (indirect ELISA).

The genes coding for nanobodies of the HB2151 *E. coli* clones that gave a positive indirect ELISA to the recombinant Wuhan RBD were subjected to Sanger sequencing (ATGC, Pathum Thani, Thailand). The DNA sequences were analyzed in CLC main workbench 21 (Qiagen, Hilden, Germany) for the presence of the *vh*/*vhh* sequences in the phagemids. The *vh*/*vhh* sequences retrieved using forward and reverse primers (R1 and R2, respectively) [31] were aligned to confirm the sequencing reactions. The *vh*/*vhh* sequences were submitted to the International ImMunoGeneTics information system^®^ server (IMGT/V-quest) to align with the human Ig of the database [32,33]. The V_H_H hallmark in the FR2—e.g., (F/Y)42, E49, R50, and (G/F)52—of the deduced amino acids was searched.

The nanobodies of the phage-infected *E. coli* clones that bound to the Wuhan RBD were preliminarily screened for their ability to neutralize SARS-CoV-2 infectivity. Nanobodies of the *E. coli* clones of interest were selected for further study.

### 2.4. Preparation of Lysates of the vh/vhh-Phagemid-Transformed HB2151 E. coli

The *vh*/*vhh*-phagemid-transformed HB2151 *E. coli* clones were grown in 10 mL of LB broth at 37 °C with shaking aeration (250 rpm) overnight. After centrifugation (10,000× *g*, 4 °C for 5 min), the bacteria in each pellet were resuspended in 1 mL of PBS and subjected to sonication for 90 s (30% power, 5 s pulse-on and 1 s pulse-off). The preparation was centrifuged (15,000× *g*, 4 °C for 5 min), and the supernatants (*E. coli* lysates) were collected. Alternatively, lysates of the *E. coli* were prepared by adding 500 µL of 1× BugBuster^®^ protein extraction reagent (Millipore, Merck KGaA, Darmstadt, Germany) to the bacterial pellet derived from 10 mL of log-phase-grown culture. After centrifugation (15,000× *g*, 4 °C for 5 min), the *E. coli* lysates (supernatants) were collected.

### 2.5. Large-Scale Production of the RBD-Bound Nanobodies

The *vh*/*vhh*-pCANTAB 5E phagemids were isolated from the HB2151 *E. coli* clones by using the FavorPrep plasmid extraction mini kit (Favorgen, Wien, Austria). The concentrations of the isolated plasmids were measured (NanoDrop8000, Thermo Fisher Scientific). Forty micrograms of the isolated plasmids from individual *E. coli* clones was cut with *Not*I and *Sfi*I restriction endonucleases (FastDigest, Thermo Fisher Scientific) and ligated to the similarly cut pET23b+ vector backbone by using T4 ligase (Thermo Fisher Scientific). The ligated products were introduced to the JM109 *E. coli* cloning host (K12, endonuclease-deficient; for improving the stability and quality of the miniprep-inserted DNA) using the TransformAid bacterial transformation kit (Thermo Fisher Scientific). The transformed JM109 *E. coli* clones were grown in LB broth supplemented with 100 µg/mL ampicillin (LB-A broth) overnight. The plasmid DNAs were extracted and Sanger-sequenced. The plasmids with verified *vh*/*vhh* sequences were introduced to NiCo21 (DE3) *E. coli* (an engineered BL21 (DE3) *E. coli* in which endogenous metal-binding proteins have been minimized, increasing the purity of the polyhistidine-tagged recombinant protein after isolation by immobilized metal affinity chromatography (IMAC)). For expression of the nanobodies (VHs/V_H_Hs), the selected NiCo21 (DE3) *E. coli* clones were grown in LB-A broth overnight; 1% of each starter culture was inoculated into 250 mL of fresh LB-A broth and grown at 37 °C with shaking aeration (250 rpm) for 3 h. Isopropyl β-d-1-thiogalactopyranosid (IPTG) was added to a 1 mM final concentration, and the preparations were incubated further at 30 °C for 4 h. The IPTG-induced bacteria were collected by centrifugation (8000× *g*, 4 °C for 15 min), and the bacterial inclusion bodies (IBs) were isolated by a non-chromatographic method [34]. Briefly, each gram of the bacterial paste was lysed in 5 mL of BugBuster^®^ protein extraction reagent containing 10 μL of Lysonase (Millipore). The preparations were centrifuged (15,000× *g*, 4 °C, 10 min), and the IBs in the pellets were washed with 10 mL of wash-100 buffer (phosphate buffer, pH 8.0; 500 mM sodium chloride (KemAus, New South Wales, Australia); 5 mM ethylenediaminetetraacetic acid (KemAus); and 1% (*v*/*v*) Triton X-100 (Affymetrix, Thermo Fisher Scientific)), 10 mL of wash-buffer-114 (phosphate buffer, pH 8.0; 50 mM sodium chloride; and 1% (*v*/*v*) Triton X-114 (Sigma, St. Louis, MO, USA, Merck KGaA)), and 10 mL of deionized distilled water, respectively. The IBs were then solubilized in 1 mL of solubilization buffer (50 mM CAPS, pH 11.0 (Sigma, Merck KGaA), supplemented with 0.3% (*w*/*v*) sodium lauroyl sarcosinate (Sigma, Merck KGaA) and 1 mM dithiothreitol (DTT, Affymetrix, Thermo Fisher Scientific)). After removing the insoluble portion by centrifugation as described above, the solubilized recombinant 6× His-tagged nanobodies were refolded in 20 mM Tris, pH 8.5, with and without 0.1 mM DTT. The refolded nanobodies were checked by SDS-PAGE, Coomassie Brilliant Blue G-250 (CBB) staining, and Western blotting.

### 2.6. Indirect Enzyme-Linked Immunosorbent Assay

Indirect ELISA was performed for checking the binding of the E-tagged nanobodies in lysates of *vh*/*vhh*-phage-transformed HB2151 *E. coli* to the homologous RBD. One microgram of the recombinant RBD and one microgram of BSA (control antigen) in 100 μL of PBS were added to separate wells of the 96-well MaxiSorp immunoplate (Nunc, Thermo Fisher Scientific) and kept at 4 °C overnight. The coated wells were then washed with PBS-T; each well’s surface was blocked by using protein-free blocking solution (Pierce™ Protein-Free (TBS-T) Blocking Buffer, Thermo Fisher Scientific) for 1 h. After washing with TBS-T, 100 µL of the HB2151 *E. coli* lysates was added to both RBD-coated and BSA-coated wells, and the plate was kept at 4 °C for 16 h. After washing with the TBS-T, 100 µL of rabbit anti-E-tag (ab3397, Abcam, at 1:3000 in TBS-T) was added and kept at 37 °C for 1 h. All wells were washed with TBS-T. A secondary antibody—i.e., horseradish peroxidase (HRP)-conjugated goat anti-rabbit IgG (Southern Biotech, Birmingham, AL, USA) (100 µL of 1:3000 in TBS-T)—was added to each well, and the plate was kept for an additional 1 h. The wells were washed with TBS-T, and the color was developed by adding 100 μL of ABTS (SeraCare, Milford, MA, USA). The optical density (OD) at 405 nm of all wells was determined against blanks (wells with PBS instead of the *E. coli* lysate) by using Multi Plate Reader Synergy H1 (BioTek, Santa Clara, CA, USA). Antigen-coated wells supplemented with the lysate of the original HB2151 *E. coli* (no nanobody) served as background binding controls. Positive ELISA was OD 405 nm ≥ 0.5 against the blank and above the BSA control; the original HB2151 lysate (background control) gave negative results.

For testing the binding of the purified 6× His-tagged nanobodies (produced by *vh*/*vhh*-pET23-transformed-NiCo21 (DE3) *E. coli*) to the recombinant RBD of the SARS-CoV-2 Wuhan wildtype (homologous) and Delta and Omicron variants (heterologous), 0.5 μg of individual RBDs (Fapon Biotech) and BSA were used to coat separate wells of the MaxiSorp immunoplate (Nunc, Thermo Fisher Scientific) and kept at 4 °C overnight. The coated wells were blocked with 5% skimmed milk in PBS-T at 37 °C for 1 h, washed with PBS-T, supplemented with 0.3 μg of 6× His-tagged nanobodies (20 pM) from individual *E. coli* clones (antigen:antibody molar ratio = 1:1), and incubated at 37 °C for 1 h. The wells were washed, supplemented with rabbit anti-His antibody (Abcam), and kept for a further 1 h. Goat anti-rabbit Ig-alkaline phosphatase (AP) conjugate (1:3000; Southern Biotech) was used as a secondary antibody. Para-nitrophenyl phosphate (pNPP) ELISA substrate (Stem Cell Technologies, Vancouver, BC, Canada) was used for color development at room temperature for 2 h in darkness. Optical density at 450 nm was determined against blanks (wells supplemented with PBS instead of nanobodies) by using the Multi Plate Reader Synergy H1 (BioTek).

The half-maximal effective concentration (EC50) of the nanobodies was determined against the recombinant S1 subunit (rS1) of the SARS-CoV-2 Wuhan wildtype. Wells of the ELISA plate (Nunc, Thermo Fisher Scientific) were coated individually with 0.5 μg of rS1 in 100 μL of PBS, blocked with 5% skimmed milk (PanReac AppliChem, Darmstadt, Germany) in PBS-T, and the blocked wells were supplemented with varied concentrations of nanobodies from different clones. The ELISA was developed by using rabbit anti-His (Abcam), which detected the 6× His-tagged nanobodies, goat anti-rabbit Ig-HRP conjugate, and ABTS substrate, washing with PBS-T between the steps. The optical density at 405 nm of each well was determined against blanks (wells supplemented with PBS instead of nanobodies). The EC50 of the nanobodies was calculated using nonlinear regression tests (Prism version 9.3, GraphPad Software, LLC, San Diego, CA, USA).

### 2.7. Sodium Dodecyl Sulfate–Polyacrylamide Gel Electrophoresis (SDS-PAGE) and Western Blot Analysis

Sodium dodecyl sulfate–polyacrylamide gel electrophoresis and Western blotting were performed as described previously [34]. The SDS-PAGE-separated antigens in the gels (4% stacking gel and 12% separating gel) were transblotted onto nitrocellulose membranes (Cytiva, Marlborough, MA, USA), and empty sites on the membranes were blocked with 5% (*w*/*v*) skimmed milk (PanReac AppliChem) in Tris-buffered saline containing 0.05% Tween 20 (TBS-T) at room temperature for 1 h. After blocking, the primary antibody (rabbit anti-E-tag or rabbit anti-His tag (Abcam; 1:3000 in TBS-T)) was added to the membranes for 1 h, and then the membranes were washed with TBS-T and incubated with AP-conjugated goat anti-rabbit IgG (Southern Biotech) (1:3000 in TBS-T) for 1 h. BCIP/NBT substrate (SeraCare) was used to detect reactive bands of nanobody–anti-tag complexes.

### 2.8. Confocal Microscopy

Vero E6 cells (5 × 10^4^ cells) were seeded on 8-chamber cell imaging glass cover slips (Eppendorf, Hamburg, Germany) and maintained in 2%-FBS-supplemented DMEM at 37 °C in a 5% CO_2_ incubator for 12 h. The cells were then infected with 50 pfu of SARS-CoV-2 (Wuhan, Delta, or Omicron) for 48 h; cells in medium alone were also included in the experiment. The infected and uninfected cells were washed with PBS before fixing with 10% formalin solution for 2 h. The fixed cells were washed with PBS three times and permeabilized with 0.1% Triton X-100 in PBS at room temperature for 15 min. After washing with PBS and blocking with 3% BSA in PBS at room temperature for 1 h, the cells were washed and stained separately with 6× His-tagged nanobodies to the SARS-CoV-2 RBD (20 µg/mL) at 4 °C for 1 h. After washing, the cells were supplemented with mouse anti-6× His antibody (Invitrogen) and rabbit anti-S1 subunit of SARS-CoV-2 spike protein (S1 mAb, Cat. 99423S; Cell signaling, Danvers, MA, USA) at dilutions of 1:200 and 1:300, respectively, for 1 h and washed. Secondary antibodies, including 1:400 dilutions of Alexa Fluor Plus 488 goat anti-rabbit IgG (Invitrogen) and Alexa Fluor Plus 555 goat anti-mouse IgG (Invitrogen), were added to the cells and kept at 4 °C for 1 h. The cells were washed with PBS, and their nuclei were stained with DAPI (Invitrogen) for 10 min. After washing, the cover slips were mounted, and the cells were examined using laser scanning confocal microscopy (Nikon C2+ Eclipse Ti2-E Laser Confocal Microscope, Nikon, Melville, NY, USA).

### 2.9. Nanobody-Mediated Neutralization/Enhancement of SARS-CoV-2 Infectivity

Vero E6 cells (1.5 × 10^5^ cells) were seeded in 24-well cell culture plates and incubated at 37 °C in a 5% CO_2_ incubator overnight. Nanobodies (0.25, 0.5, 1.0, 1.5, and 2.0 mM) or medium alone (negative control) were mixed with 50 pfu of SARS-CoV-2 (Wuhan, Delta, or Omicron) and incubated at 37 °C for 1 h. The mixtures were added to appropriate wells containing the Vero E6 cells and incubated at 37 °C for 1 h; the fluids in all wells were discarded, and the cells were washed once with PBS. After washing, 1.5% carboxymethyl cellulose (CMC) (Sigma Aldrich, St. Louis, MO, USA, Merck KGaA) in complete DMEM was added to each well, and the plates were incubated for a further 5 days. The numbers of plaques formed in the infected cell monolayers in all wells were determined by using a plaque-forming assay.

### 2.10. Plaque-Forming Assay (PFA)

SARS-CoV-2-infected cells in the culture wells were fixed with 10% formaldehyde at room temperature for 2 h, washed with distilled water five times to get rid of the CMC, and stained with 1% crystal violet in 10% ethanol at room temperature for 15 min. After washing with distilled water, the plates were dried, and plaques were counted visually. The plaque numbers were compared between treatment groups.

### 2.11. Competitive ELISA for Testing Binding of Neutralizing Nanobodies to the RBD of SARS-CoV-2’s Wuhan Wildtype, Delta, and Omicron Variants

Nanobodies that neutralize SARS-CoV-2’s infectivity were tested as to whether they could interfere with the RBD–human ACE2 interaction. Nanobodies (25 μg in 60 μL of PBS) were mixed with HRP-conjugated RBDs of SARS-CoV-2’s Wuhan wildtype strain and the Delta and Omicron variants (Fapon Biotech) (300 μg in 60 μL of PBS), and they were kept at 37 °C for 30 min. Biotin-conjugated human ACE2 (Fapon Biotech) was diluted 1:800 with PBS, and 100 μL was added to each well of a streptavidin microplate (Fapon Biotech); the plate was incubated at 37 °C for 30 min, and all wells were then washed three times with PBS-T. The nanobody–RBD mixtures were added to appropriate ACE2-coated wells. Positive (human anti-SARS-CoV-2 RBD, Fapon Biotech) and negative controls (PBS-T) were also included in the assay. The plate was incubated at 37 °C for 15 min; the fluid in each well was discarded, and all wells were washed three times with PBS-T. TMB (3,3′,5,5′-tetramethylbenzidine) substrate (KPL, Milford, MA, USA) was added to each well (100 μL per well), and the plate was kept at 25 °C in darkness for 20 min. The enzymatic reaction was stopped by adding 50 μL of 1 N sulfuric acid to each well. The optical density at 450 nm of all wells was determined immediately using a microplate reader (Hercuvan, Cambridge, UK). The percentages of inhibition of RBD binding to immobilized hACE2 mediated by the nanobodies and human immune serum were calculated in comparison to the negative inhibition (PBS-T) controls.

### 2.12. Identification of the Nanobody-Bound Epitopes by Phage Mimotope Search and Multiple Sequence Alignment

Phage-displayed peptides that were bound to the nanobodies (phage mimotopes) were searched and used in multiple sequence alignment with SARS-CoV-2 RBD linear sequences for identification of the regions of the SARS-CoV-2 RBD that were bound by the nanobodies (i.e., presumptive epitopes of the nanobodies). For the phage mimotope identification, 1 μg of purified nanobodies of individual *E. coli* clones in 100 µL of PBS was added to separate wells of a MaxiSorp immunoplate (Nunc, Thermo Fisher Scientific) and kept at 4 °C overnight. The nanobody-coated wells were washed with TBS-T, blocked with protein-free blocking solution (Pierce™ Protein-Free Blocking Buffer, Thermo Fisher Scientific) for 1 h, and washed again with TBS-T. Ten microliters of the Ph.D.™-12 phage display peptide library (New England BioLabs, Ipswich, MA, USA) containing ~10^11^ 12-mer-peptide-displaying phage particles in 100 µL of TBS-T was added to each nanobody-coated well. The phages were allowed to bind to the nanobodies at room temperature for 1 h. The unbound phages were removed by washing them away with TBS-T; the nanobody-bound phages were eluted with 100 µL of 0.2 M glycine-HCl (pH 2.2) containing 0.1% BSA and immediately supplemented with 15 µL of 1 M Tris-HCl (pH 9.6). The eluted phages were mixed with 20 mL of early-log-phase-grown ER2738 *E. coli* (an amber suppressor (glnV) F^+^ strain), and incubated at 37 °C with shaking (250 rpm) for 4.5 h. After centrifugation (11,000× *g*, 4 °C, 15 min), the culture supernatants containing phage particles were collected and precipitated by adding a 1/6 volume of 20% polyethylene glycol 8000 (PEG8000) in 2.5 M NaCl at 4 °C overnight, and then centrifuged (11,000× *g*, 4 °C, 15 min). The phage pellet was resuspended in TBS and used in the next round of panning with the same nanobodies. Three panning rounds were performed. The phages of the third panning round were diluted 10-fold serially, and 10 µL of each dilution was used to infect 200 µL of mid-log-phase-grown ER2738 *E. coli* for 5 min. The phage-infected ER2738 *E. coli* bacteria were mixed with 3 mL of 45 °C molten LB agar and poured onto LB-IPTG-X-gal agar plates. The plates were incubated at 37 °C overnight. The blue colonies on the agar plates (within hundred-ranged blue colonies) were counted. The phage concentration was then calculated as plaque-forming units (pfu). Twenty isolated blue colonies of ER2738 *E. coli* were picked and grown. The phagemids were isolated from individual *E. coli* cultures by the phenol/chloroform method and Sanger-sequenced. The 12-mer-peptide sequences were deduced from the respective DNA sequences. The peptides displayed by the 20 nanobody-bound phage clones (designated phage mimotope (M) types 1–20) were then aligned (Clustal Omega multiple sequence alignment) with the RBD sequences of SARS-CoV-2’s Wuhan wildtype and variants to locate the RBD regions and residues bound by the respective nanobodies (epitopes of the nanobodies).

### 2.13. Peptide-Binding ELISA

Peptide-binding ELISA was performed for validation of the nanobody-presumptive epitopes that were identified indirectly by using phage mimotopes and multiple RBD sequence alignment. For this experiment, the biotin-labelled RBD consensus peptides that matched the phage mimotopes were synthesized (GenScript, Piscataway, NJ, USA) and used as antigens in the indirect ELISA for testing the direct binding of the nanobodies. A control (irrelevant) peptide was included in the experiment for negative binding. Individual peptides (1 μg in 100 μL of PBS) were used to coat streptavidin microplates (Thermo Fisher Scientific) in triplicate wells at 4 °C overnight. The antigen-coated wells were washed with PBS-T and blocked with 5% skimmed milk (PanReac AppliChem) before adding 1 μg of nanobodies. The plates were incubated at 37 °C for 1 h, washed with PBS-T, supplemented with rabbit anti-His antibody (Abcam), and incubated for 1 h. After washing with PBS-T, all wells were supplemented with anti-rabbit Ig-HRP conjugate (Southern Biotech). ABTS (2,2’-azino-bis (3-ethylbenzothiazoline-6-sulfonic acid)) chromogenic substrate (KPL) was used for color development. The optical density at 405 nm of each well was determined against blanks (wells supplemented with PBS instead of antibodies).

### 2.14. Computerized Simulation for Determining Residues and Regions of the SARS-CoV-2 RBD Bound by the Nanobodies

The deduced amino acid sequences of the nanobodies were submitted to ColabFold notebook for Deepmind’s AlphaFold2-based three-dimensional (3D) structure building [35]. The Jackhmmer method was used to create multiple sequence alignment (MSA). The predicted nanobody models were refined by the built-in Amber-Relax tool. The SARS-CoV-2 RBD (RCSB PDB: 7WK2) on the surface spike (S) glycoprotein crystal structure and the nanobody 3D models were submitted to the HADDOCK server to determine protein–antibody interaction using expert mode [36]. All CDRs of each nanobody were set as active residues. The number of structures for rigid body docking was set to 10,000, the number of structures for semi-flexible refinement was set to 400, the number of structures for the final refinement was set to 400, and the number of structures to analyze was set to 400. The models with the highest HADDOCK scores were chosen for detailed analysis. The protein structural models and their molecular interactions were built and visualized using PyMOL software (Schrödinger, LLC, New York City, NY, USA).

### 2.15. Statistical Analysis

Prism version 9.3 (GraphPad Software, LLC, San Diego, CA, USA) was used to compare the results of the tests and controls. Statistically significant differences were determined by one-way ANOVA and the Bonferroni test; *p*-values < 0.05 were considered statistically significant: *, *p* < 0.05; **, *p* < 0.01; ***, *p* < 0.001; ****, *p* < 0.0001.

## 3. Results

### 3.1. Nanobodies to the SARS-CoV-2 Receptor-Binding Domain

The phages displaying nanobodies (VHs/V_H_Hs) that bound to the RBD of SARS-CoV-2’s Wuhan wildtype from the phage panning were used to infect HB2151 *E. coli*. The phage-infected HB2151 *E. coli* preparations were spread on selective LB-A agar plates. After overnight incubation at 37 °C, there were 190 colonies of phage-transformed *E. coli* that appeared on the agar. These *E. coli* clones were screened for nanobody-coding genes (*vhs*/*vhhs*) by direct colony PCR using oligonucleotide primers (R1 and R2) specific to the recombinant pCANTAB 5E phagemids. Among the 190 phage-infected clones, 89 clones carried the inserted DNAs of ~600 bp, which is the correct size of *vh*/*vhh* amplicons with the phage DNA-flanking regions. The amplicons from representative phage-infected HB2151 *E. coli* clones are shown in Figure 1A. The *vh*/*vhh*-positive *E. coli* clones were grown in LB broth under IPTG-induced conditions, and their lysates were tested for the presence of E-tagged nanobodies (VHs/V_H_Hs) by Western blotting, using an anti-E tag antibody as the tracer. Of the 89 *vh*/*vhh*-positive clones, 44 clones expressed recombinant proteins at 17–20 kDa, which are the molecular sizes of the nanobodies; representatives are shown in Figure 1B. The nanobodies in the lysates of these 44 clones were then tested for binding to the homologous Wuhan RBD by indirect ELISA, and 16 clones (no. 103, 105, 114, 148, 156, 160, 162, 184, 187, 215, 219, 228, 256, 265, 278, and 285) gave positive results (Figure 1C).

The *vhs*/*vhhs* of the 16 HB2151 *E. coli* clones were Sanger-sequenced, deduced to aa sequences, and submitted to the IMGT V-Quest server to determine their similarity to human immunoglobulin framework regions (FRs). The V-region similarity of the nanobodies (VHs/V_H_Hs) to the human VHs of the database ranged from 81.79 to 98.96%, with the median at 91.58% (Table 1). The length of the CDR3 of the human-like nanobodies ranged between 8 and 22 amino acid residues, with the mode at 22 residues (Table 1). The FR2 of clone 219 revealed the characteristic amino acid tetrad of V_H_H, i.e., (F/Y)(42)E(49)R(50)(G/F)(52), which was different from the FR2 of the conventional VH, which contains V(42)G(49)L(50)W(52).

### 3.2. Preliminary Screening of Nanobodies to the Wuhan RBD against SARS-CoV-2 Infectivity

Nanobodies that bound to the Wuhan RBD in lysates of the 16 phage-infected HB2151 *E. coli* clones were initially screened for their ability to neutralize the infectivity of SARS-CoV-2 (Wuhan and Delta) in infected Vero E6 cells. The results of this initial screening revealed that among the 16 clones, nanobodies of clones 114 and 278 (VH114 and VH278) showed high neutralizing activity against both SARS-CoV-2 strains (at the time of this initial screening, the Omicron isolate was not available). Unexpectedly, the nanobodies in the lysates of phage-transformed HB2151 *E. coli* clones 103 and 105 (VH103 and VH105) enhanced the infectivity of both SARS-CoV-2 strains by increasing the plaque numbers in the infected Vero E6 monolayer (data not shown) compared to the infected cells in medium alone. From this preliminary experiment, the nanobodies of the HB2151 *E. coli* clones 103, 105, 114, and 278 (E-tagged- VH103, VH105, VH114, and VH278, respectively) were produced at large scale in the form of 6× His-tagged nanobodies and investigated further. The remaining clones were kept aside for later study.

### 3.3. Nanobodies to the Wuhan RBD Also Bound to Recombinant RBDs of the Delta and Omicron Variants and the Native S1 Subunit of the SARS-CoV-2 Spike Protein

After subcloning the *vhs* coding for Wuhan RBD-bound VH103, VH105, VH114, and VH278 from pCANTAB 5E phagemids to pET23b+ plasmids, the nanobodies expressed by the *vh*-pET23b+ plasmids transformed NiCo21 (DE3) *E. coli*, i.e., the 6× His-tagged VH103, VH105, VH114, and VH278 were purified (Figure 2A). These nanobodies also bound to recombinant RBDs of the Delta and Omicron variants, as tested by indirect ELISA using BSA as a control antigen (Figure 2B). The 6× His-tagged VH103, VH105, VH114, and VH278 that bound to recombinant RBDs of the Wuhan wildtype and Delta and Omicron variants also bound to the native S proteins in SARS-CoV-2-infected Vero E6 cells, as determined by confocal microscopy (Figure 2C–E). The half-maximal effective concentration (EC50) of the enhancing VH103 and VH105 against the recombinant S1 subunit was 195 and 121 nM, respectively—more than those of VH114 (14.54 nM) and VH278 (18.97 nM) (Figure 2F).

### 3.4. Nanobody-Mediated Neutralization/Enhancement of SARS-CoV-2 Infectivity

The numbers of plaques formed in the Vero E6 monolayer supplemented with SARS-CoV-2 (Wuhan, Delta, and Omicron) that had been incubated with different concentrations of the nanobodies VH103, VH105, VH114, and VH278 (0.25, 0.5. 1.0, 1.5, and 2.0 μM) or in medium alone (0 μM nanobodies) for 5 days (each condition in triplicate) were determined by using a plaque-forming assay (PFA). Representative results of each of the three reproducible and independent neutralization/enhancement assays are shown in Figure 3A–C (unprocessed images are provided as Appendix A). VH103 and VH105 enhanced the viral infectivity in a dose-dependent manner, while VH114 and VH278 dose-dependently neutralized the infectivity of the SARS-CoV-2 viruses (Figure 3D–F and Table 2). Statistical comparisons of the neutralization/enhancing activities of the four nanobodies at the same concentrations are shown in Appendix A.

### 3.5. Neutralizing Nanobodies Inhibited Binding of the SARS-CoV-2 RBDs to Human ACE2

The results of the competitive ELISA revealed that the VH114 and VH278 could inhibit binding of the Wuhan, Delta, and Omicron RBDs to the immobilized human ACE2 (Figure 4). At 25 µg, VH114 inhibited the binding of the Wuhan, Delta, and Omicron RBDs (300 µg) to hACE2 by 76.4, 56.3, and 79.8%, respectively, while VH278 inhibited the same RBD–hACE2 interactions by 79.1, 51.3, and 77.2%, respectively. The interactions were inhibited by the human anti-SARS-CoV-2 RBD (positive control) at 97.2, 98.1, and 83.7%, respectively. The background inhibition of the negative controls was 0.6, 2.9, and 0.2%, respectively.

### 3.6. Phage Mimotope Search and Alignments of the Phage Mimotopic Peptides with RBD Sequences

Presumptive epitopes of the enhancing (VH103 and VH105) and neutralizing (VH114 and VH278) nanobodies were identified indirectly by means of phage mimotope search and alignment of the nanobody-bound phage peptide sequences with linear RBD sequences of multiple SARS-CoV-2 variants. A diagram of the domain organization of the S1 subunit of the SARS-CoV-2 spike (S) protein [10] is shown in Figure 5. The Ph.D.^TM^-12 phage display peptide library was used as a tool to identify sequences of peptides that bound to the enhancing and neutralizing nanobodies. After three rounds of phage panning with individual nanobodies, 20 nanobody-bound phage clones were randomly picked for phage DNA extraction and sequencing. The inserted DNAs in the recombinant phage genomes were deduced, and the deduced peptides were designated mimotope (M) types 1–20. Each M type was then aligned with monomeric RBD sequences of different SARS-CoV-2 strains—including Wuhan wildtype, Delta (B.1.617.2), and Omicron (B.1.1.529/BA1, BA2, BA4, and BA5)—by using Clustal Omega multiple sequence alignment for identification of the RBD regions bound by the nanobodies, i.e., presumptive epitopes of the nanobodies. The sequences of the phage mimotopes (M types), the regions of the S1 RBD that matched the phage mimotopic peptides, and the consensus RBD sequences are shown in Figure 5A,B.

There was a cluster of 11 M types (M3, M4, M6-M8, M10, and M15-M19) from phage panning with the enhancing VH103 nanobody that matched the linear peptides of SARS-CoV-2 S1 RBD residues, i.e., aa 354–381 (Wuhan), 357–379 (Delta B.1.617.2), 356–378 (Omicron B.1.1.529/BA1 and BA2), and 354–376 (Omicron BA4 and BA5) (left panel of Figure 5A). The consensus RBD peptides that contained the VH103 presumptive epitope were Wuhan and Delta variant 359/357NCVADVSVLYNSASFSTFKCYG381/379 (designated peptide VH103-P1), Omicron BA1 variant 356NCVADVSVLYNLAPFFTFKCYG378 (designated peptide 103-P2), and Omicron BA2, BA4, and BA5 356/354/354NCVADVSVLYNFASFSTFKCYG378/376/376 (designated peptide VH103-P3). This VH103-bound region is outside the receptor-binding motif (RBM, Wuhan RBD aa 437–508).

There were seven M types (M1, M4, M5, M8, M9, M15, and M11) that bound to the enhancing VH105 nanobody (right panel of Figure 5A). Based on the Wuhan wildtype, the M1 mimotopic peptide matched residue 404GDEVRQIAPGQT415; M4, M9, and M15 displayed the same peptide that matched residue 423YKLPDDFTGCVI434; the M5 matched residue 385TKLNDLCFTNVY396; the M8 matched residue 407VRQIAPGQTGKI418; and the M11 matched residue 420DYNKLPDDFTGC431. The epitope of the enhancing VH105 is also located outside the RBM. The RBD consensus peptides of M5 (TKLNDLCFTNVY), M1+M8 (GDEVRQIAPGQTGNI), and M4/M9/M15+M11 (DYNKLPDDFTGCVI) were designated VH105-P4, VH105-P5, and VH105-P6, respectively.

Ten mimotope (M) types of neutralizing VH114-bound phages (M8-15, M18, and M20) matched S1 RBD residues 340–357 (Wuhan), 338–355 (Delta B.1.617.2), 357–354 (Omicron B.1.1.529/BA1 and BA2), and 335–352 (Omicron BA4 and BA5) of SARS-CoV-2 S1 subunits. The RBD consensus sequence 340EVFNATRFASVYAWNRKRI357 that matched these 10 M types was also located outside the RBM (right panel of Figure 5B). The sequence EVFNATRFASVYAWNRKRI was conserved for all aligned SARS-CoV-2 strains; this peptide was designated VH114-P7.

The neutralizing VH278 bound to a group of 13 phage M types that matched aa residues in the RBD N-terminal outside the RBM, i.e., residues 319–334 (Wuhan), 317–332 (Delta B.1.617.2), 316–331 (Omicron B.1.1.529/BA1 and BA2). and 314–329 (Omicron BA4 and BA5) (left panel of Figure 5B). The consensus sequence was 319-RVQPTESIVRFPNITN334. This peptide (RVQPTESIVRFPNITN, designated VH278-P8) was conserved across all SARS-CoV-2 strains that were used in the multiple alignment.

The other phage peptides did not match the RBD sequences, or the phage clones did not contain peptide-coding genes.

### 3.7. Peptide-Binding ELISA for Validation of the Presumptive Epitopes

The consensus peptides of the SARS-CoV-2 RBD that contained presumptive epitopes of the nanobodies VH103, VH105, VH114, and VH278—including VH103-P1, VH103-P2, and VH103-P3; VH105-P4, VH105-P5, and VH105-P6; VH114-P7; and VH278-P8, respectively—were synthesized commercially (GenScript) and used as the antigens in the peptide-binding ELISA for testing the binding of the nanobodies. As shown in Figure 6, the nanobodies bound to their respective peptides, yielding significantly higher ELISA signals compared to the control peptide. The VH103 bound to VH103-P1, VH103-P2, and VH103-P3; the VH105 bound to VH105-P4, VH105-P5, and VH105-P6; the VH114 bound to VH114-P7; and the VH278 bound to VH278-P8; all nanobodies gave negligible binding signals to the control peptide, indicating that the tested peptides contained epitopes/parts of the epitopes of the nanobodies, which validates the results obtained by the mimotope identification and multiple RBD sequence alignment.

### 3.8. Computerized Simulation to Identify Residues and Regions of the SARS-CoV-2 RBD Bound by the Nanobodies

To investigate the interaction of the nanobodies with the RBDs of SARS-CoV-2’s trimeric S protein, homology models of nanobodies of clones 103, 105, 114, and 278 were built. The VH 3D models were docked with the trimeric S protein. The results are shown in Figure 7 and Table 3. The enhancing VH103 uses its long CDR3 (21 residues) to interact with seven residues in two RBDs (RBDs 1 and 3) of the trimeric S protein; among them, four amino acids (R403, Y489, R493, and K440) are known to bind tightly to human ACE2 [37]. The enhancing VH105 uses all three CDRs to interact with 12 residues of the RBD1 and 1 residue of the RBD2; 11 of the 12 residues in the RBD1 are known to bind tightly to human ACE2 [37]. The CDR3s of the neutralizing VH114 and VH278 are relatively short (nine and eight residues, respectively) compared to other VHs (Table 1). VH114 uses all three CDRs to cooperatively form contact interfaces with nine residues of RBD1; eight amino acids are not tight ACE2-binding ones. Likewise, VH278 uses all CDRs to interact with four amino acids (non-ACE2-tight-binding residues) of the RBD2.

## 4. Discussion

Receptor recognition by viruses is a prerequisite step of viral infectivity and pathogenesis, as well as their host species determination [38,39]. Thus, interfering with the receptor recognition is a strategic basis for the development of virus vaccines and antiviral agents. For SARS-CoV-2 (and SARS-CoV), conformational changes of the receptor-binding domain (RBD) on the trimeric spike (S) protein to the open (up) position enhance RBD binding to human ACE2 (hACE2) and cell entry [5]. Blocking interaction between the RBD and hACE2 prevents cells from being infected [40]. Monoclonal antibodies (mAbs) that neutralize SARS-CoV-2’s infectivity have been generated for the treatment of COVID-19, especially for severely ill patients and for disease intervention among those infected who are refractory to vaccination, such as immunocompromised subjects and the elderly [41]. Currently, there are more than 50 anti-SARS-CoV-2 mAbs, some of which—either alone or combination—have been approved for COVID-19 treatment, while the others are at different stages of development [41]. Nevertheless, treatment or intervention of viral diseases using intact (four-chain) antibodies has limitations due to antibody-dependent enhancement (ADE) via different mechanisms (as mentioned earlier), which could eventually lead to aggravation of the tissue pathology and symptom severity [13,14,15,16,17,18,19,20,21,23,24,25,26]. Thus, intensive screening of the antibody preparations for treatment of a particular viral infection is required. In this study, single-domain antibodies (monovalent nanobodies) that are devoid of Fc fragments, cannot crosslink the target, and cannot activate complement and immune cells were generated for further development towards clinical application as relatively safe COVID-19 therapeutics.

The nanobody (VH/V_H_H)-displaying phage clones that bound to the RBD of the SARS-CoV-2 Wuhan wildtype were selected from our previously constructed nanobody (VH/V_H_H) phage display library. For constructing this library, the *vh*/*vhh* genes used for the library construction were amplified from cDNAs of camelid immunoglobulin genes by using oligonucleotide primers designed from human immunoglobulin genes; thus, only the camelid cDNA templates that were recognized by the human primers could be amplified [30]. The *vh*/*vhh* sequences in this phage library showed high homology to genes coding for human VH framework regions (FRs), as shown in Table 1 and previously reported data [30,42,43,44,45]. Therefore, the nanobodies may be regarded as human-like proteins and should have negligible—if any—immunogenicity in the human recipients if used for the treatment of COVID-19 or passive immunization.

The single-round phage panning that was successfully used previously in our laboratory was applied for the selection of RBD-bound phages in this study, as we found previously that the multiple panning rounds (as performed by other laboratories) resulted in the loss of the antibody-coding genes from the recombinant phages. Because of the single-round panning, there were several phages that did not express the nanobodies, or whose expressed nanobodies (VHs/V_H_Hs) did not bind to the panning antigen (non-specific phages). These phages were discarded, and only 16 phage clones carried nanobody genes whose expressed proteins bound specifically to the Wuhan RBD (a homologous antigen used as bait to fish out the RBD-bound phages from the library).

It is known that mutations in the RBD of SARS-CoV-2 render the virus mutants able to escape binding by different classes of antibodies (e.g., vaccine-induced, convalescing, and therapeutic mAbs) [46,47]. The Delta variant is known to cause high case and fatality rates among infected patients. This variant has two mutated residues, i.e., L452R and T478K, in the receptor-binding motif (RBM) of the RBD, as well as other mutations in the C-terminal domain of the S1 subunit outside the RBD, including D614G and P618R, and four point mutations in the S1 N-terminal domain, i.e., T19R, G142D, del156-F157, and R158G [46]. The Omicron variant contains 15 point mutations in the RBD (G339D, S371L, S373P, S375F, K417N, N440K, G446S, S477N, T478K, E484A, Q493R, G496S, Q498R, N501Y, and Y505H), 5 point mutations in the S1 C-terminal domain (T547K, D614G, H655Y, N679K, and P681H), and 7 point mutations in the S1 N-terminal domain (A97V, delH69-V70, T95I, G142D, delV143-Y145, N211I + delL212, and insR214-EPE). Many mutations that cause increased numbers of positive charges in specific regions of the S protein confer greater and stronger interactions (via Coulombic force) between the S protein and the hACE2 receptor, enhancing the viral entry [47], i.e., both the Delta and Omicron variants have higher infectivity and transmissibility than the original Wuhan and Alpha variant strains [48]. Among the 16 *vh*/*vhh* phagemid-transformed HB2151 *E. coli* clones, we selected 4 clones whose expressed nanobodies (VH103, VH105, VH114, and VH278) bound not only to the RBD of the homologous Wuhan wildtype, but also to the recombinant RBDs of the SARS-CoV-2 Delta and Omicron variants (indirect ELISA) and S1 subunits of the native spike proteins of Wuhan, Delta, and Omicron in the infected cells (confocal microscopy).

Initial screening against the Wuhan strain and the Delta variant revealed that the soluble E-tagged nanobodies in the lysates of phage-transformed HB2151 *E. coli* clones 114 and 278 (VH114 and VH278) neutralized the infectivity of both SARS-CoV-2 strains. After large-scale production and purification of 6× His-tagged nanobodies from the *vh-*pET23b+ plasmid-transformed NiCo21 (DE3) *E. coli* clones, the neutralizing activity of the two nanobodies at different concentrations was reverified against the Wuhan wildtype, Delta, and Omicron strains. The two nanobodies neutralized the viral infectivity in a dose-dependent manner. The EC50 of the nanobodies against the S1 subunits was in nanomolar range (14.54 nM for VH114 and 18.97 nM for VH278). Thus, they may be tested further towards clinical application as safe therapeutics for COVID-19, as well as for passive immunization to reduce morbidity.

Presumptive epitopes of the nanobodies identified indirectly by means of phage mimotope search and multiple sequence alignment of the phage mimotopes with the RBD sequences indicated that the epitope of the neutralizing VH114 was located in the residue 340EVFNATRFASVYAWNRKRI357 (based on the Wuhan RBD), which is outside the receptor-binding motif (RBM; Wuhan aa 437–508). The VH114 bound to consensus peptide VH114-P7 (EVFNATRFASVYAWNRKRI), verifying that the epitope of the neutralizing VH114 is a linear epitope contained in the VH114-P7 sequence. It has been reported [49] that the R345 peptide containing SARS-CoV-2 RBD aa TRFASVYAWNRKRISNCVAD reacted with immune sera of mice and swine that were immunized with recombinant SARS-CoV-2 RBD mixed with aluminum hydroxide/CpG1018 adjuvants, indicating that the 345 peptide contained an immunogenic B-cell epitope. Moreover, mouse monoclonal antibodies against BSA-conjugated R345 peptide (clone 10D2) inhibited the RBD–ACE2 interaction, with an inhibition rate of 20–40%; thus, the R345 peptide contained a neutralizing epitope [49]. The minimum binding motif within the R345 peptide was “VYAWN” [49]. This VYAWN motif is also contained in the consensus peptide bound by VH114, as identified by the phage mimotope search, sequence alignment, and peptide-binding ELISA in this study, verifying the previous notion that VYAWN is a SARS-CoV-2-neutralizing epitope. VYAWN is highly conserved among SARS-CoV-2 isolates, as shown in Figure 4B and [49]; thus, RBD peptides containing this motif (or its coding sequence) should be included in vaccines against SARS-CoV-2, along with other previously identified neutralizing epitopes—especially those that contain RBM key residues for interacting with hACE2 [37]. The neutralizing VH114 nanobody should be tested and developed further for use as a safe, passive, immunotherapeutic agent against COVID-19—preferably in a neutralizing mAb cocktail, as one epitope-specific mAb should not be sufficiently effective to block completely the S1–hACE2 interaction.

VH278 is another nanobody that neutralized the infectivity of the SARS-CoV-2 Wuhan, Delta, and Omicron strains that we tested. Epitope mapping via the phage mimotopes and multiple sequence alignment showed that the presumptive epitope of VH278 encompassed consensus residues in the N-terminal domain of the RBD, i.e., Wuhan aa 319–334, Delta B.1.617.2 aa 317–332, Omicron B.1.1.529/BA1 and BA2 aa 316–331, and Omicron BA4 and BA5 aa 314–329, which also lie outside the RBM. VH278 bound to the consensus RBD peptide VH278-P8 (RVQPTESIVRFPNITN), indicating that this peptide contains a neutralizing B-cell epitope. However, the synthetic R315 peptide containing TSNFRVQPTESIVRFPNITN did not bind to antibodies in the immune sera of mice and swine immunized with the SARS-CoV-2 RBD mixed with an adjuvant [49]. The different results of these two studies may reflect the differences in methods of antibody production and functional assays, as well as the antibody format, i.e., intact mouse/swine polyclonal antibodies [49], which might contain relatively small amounts of antibody to the R315 peptide in the immune sera compared to the highly purified monoclonal single-domain VH278 nanobody (this study). Our study indicates that the VH278 nanobody binds to a novel RBD-neutralizing epitope. Other neutralizing epitopes of the SARS-CoV-2 spike protein have been identified previously [49,50,51].

The phage mimotopes and sequence alignment identified the RBD consensus sequence 359NCVADVSVLYNSAPFFTFKCYG380 outside the RBM as the VH103 presumptive epitope. The VH103 nanobody bound to VH103-P1, VH103-P2, and VH103-P3 in the peptide-binding ELISA, as well as to native S proteins of the Wuhan, Delta, and Omicron strains under confocal microscopy, indicating that the VH103 epitope that has the potential to cause undesirable ADE is in this region of the SARS-CoV-2 spike protein. However, the R360 peptide NCVADYSVLYNSASFSTFKC [49] did not bind to mouse/swine polyclonal antibodies against the SARS-CoV-2 RBD [49]. The explanations for the unconfirmed results should be the same as for VH278 versus mouse/swine immune sera.

The phage mimotopes that were bound by enhancing VH105 matched different regions of the RBD, indicating that the epitope of enhancing VH105 may be a conformational epitope that is formed by amino acids of the three RBD regions that are spatially juxtaposed upon the protein folding.

Antibody-dependent enhancement (ADE) of viral entry is highly concerning in immunity to viral infections, in both vaccine development and antibody-based therapy. Different mechanisms of ADE have been recognized, as mentioned earlier. In this study, the enhancing activity of VH103 and VH105 nanobodies (without Fc fragments) was observed from an in vitro assay performed using infected Vero E6 cells (African green monkey kidney cells) that lacked an Fc receptor, and there were neither immune/inflammatory cells nor complements in the assays. Thus, the ADE mediated by the nanobodies cannot be extrinsic ADE (Fc-FcR- or complement-receptor-mediated). For SARS-CoV-2, it was reported that antibodies to the N-terminal domains (NTDs) of spike proteins that crosslinked the trimeric spikes on the virion surface could cause upstanding of the RBD, which increased hACE2 binding and cellular entry [20]. VH103 and VH105 are specific to the RBD, and they are monovalent; thus, they cannot crosslink the SARS-CoV-2 spike protein. Usually, the CDR3 of the antibody paratope plays a dominant role in antigen recognition and binding. The CDR3s of both VH103 and VH105 are long (21 and 22 aa, respectively); thus, they should be able to penetrate deeper and/or occupy more antigenic space than their short CDR counterparts. It is possible that the large target binding area and penetration of VH103 and VH105 triggered conformational changes of the RBD/spike protein and promoted their interaction with the receptors, which consequently increased cellular entry. It was observed for MERS-CoV that antibody binding triggered conformational changes of the spike proteins of virus-like particles (VLPs) and allowed the proteins to be cleaved by trypsin at the S2′ site, enhancing the VLP entry [21]. Our speculation needs further experimental verification.

## 5. Conclusions

Engineered nanobodies (VHs) to the SARS-CoV-2 receptor-binding domain (RBD) were generated. The nanobodies showed 81.79–98.96% framework similarity to human antibodies; thus, they may be regarded as human nanobodies. Nanobodies of two phage-infected *E. coli* clones 114 and 278 (VH114 and VH278) neutralized SARS-CoV-2 infectivity in a dose-dependent manner, while nanobodies of clones 103 and 105 (VH103 and VH105) enhanced the viral infectivity by increasing the CPE in an infected Vero E6 cell monolayer, also dose-dependently. These nanobodies also bound to recombinant Delta and Omicron RBDs and native spike proteins of the Wuhan, Delta, and Omicron variants. The epitope of the neutralizing VH114 contains the previously reported VYAWN motif, while the epitope of the neutralizing VH278 is a novel linear epitope located at 319RVQPTESIVRFPNITN334, based on the sequence of the Wuhan wildtype strain. The epitope of the enhancing VH103 is linear at Wuhan 359NCVADVSVLYNSAPFFTFKCYG380, and the epitope of enhancing VH105 is most likely conformational. The four identified B-cell epitopes of the SARS-CoV-2 RBD were highly conserved across the virus variants. This study reports for the first time the enhancing epitopes of the SARS-CoV-2 spike RBD, as well as a novel neutralizing epitope located outside the RBM. This information should be useful for the rational design of subunit SARS-CoV-2 vaccines that should contain only neutralizing epitopes (i.e., devoid of enhancing epitopes). The neutralizing nanobodies should be tested further for the treatment of COVID-19.

## Figures and Tables

**Figure 1 viruses-15-01252-f001:**
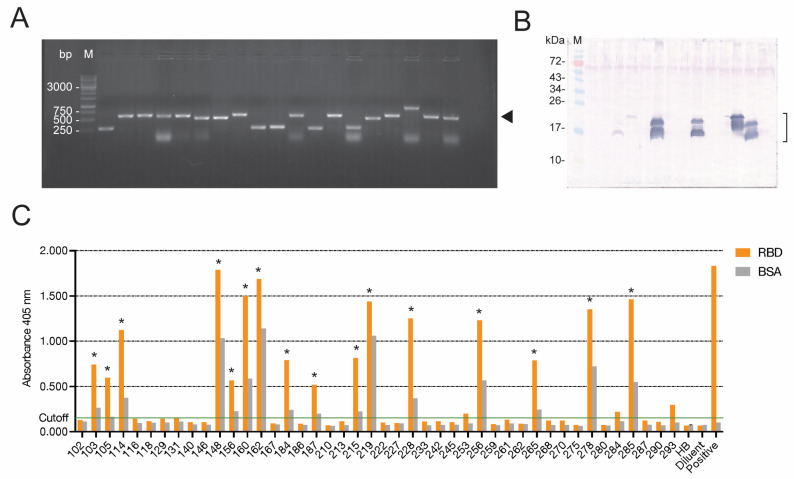
Production of nanobodies (VHs/V_H_Hs) to the SARS-CoV-2 receptor-binding domain (RBD): (**A**) Amplicons of nanobody genes (*vhs*/*vhhs)* amplified from representative phagemid-transformed HB2151 *E. coli* clones. The amplicons of the *vhs*/*vhhs* including phagemid-flanking regions are ~600 base pairs (bp) (arrowhead). M, Standard DNA ladder. The numbers on the left are DNA sizes in base pairs (bp). (**B**) E-tagged-nanobodies in the lysates of representative *vh*/*vhh*-positive HB2151 *E. coli* clones, as revealed by Western blot analysis. M, Protein molecular mass standard. The numbers on the left are protein masses in kilodaltons (kDa). The nanobodies (VHs/V_H_Hs) appeared as protein doublets at ~17–20 kDa (bracket). The upper bands are immature nanobodies (VHs/V_H_Hs) with signal peptides; the lower bands are mature proteins; other faint bands are degraded products of the principal proteins. (**C**) Indirect ELISA OD at 405 nm from binding of nanobodies in lysates of the phage-infected HB2151 *E. coli* clones to the immobilized homologous Wuhan RBD, compared to the control antigen (BSA). HB and diluent served as negative nanobody controls. HB: Wuhan-RBD-coated wells supplemented with lysate of original HB2151 *E. coli*. Diluent: Wuhan-RBD-coated wells supplemented with buffer instead of nanobodies. Positive: Wuhan-RBD-coated wells supplemented with human anti-RBD antibody (Fapon Biotech). * indicates the *E. coli* clone whose expressed nanobodies gave positive indirect ELISA results (i.e., the OD at 405 nm to the RBD was ≥0.5 and higher than the OD at 405 nm to BSA).

**Figure 2 viruses-15-01252-f002:**
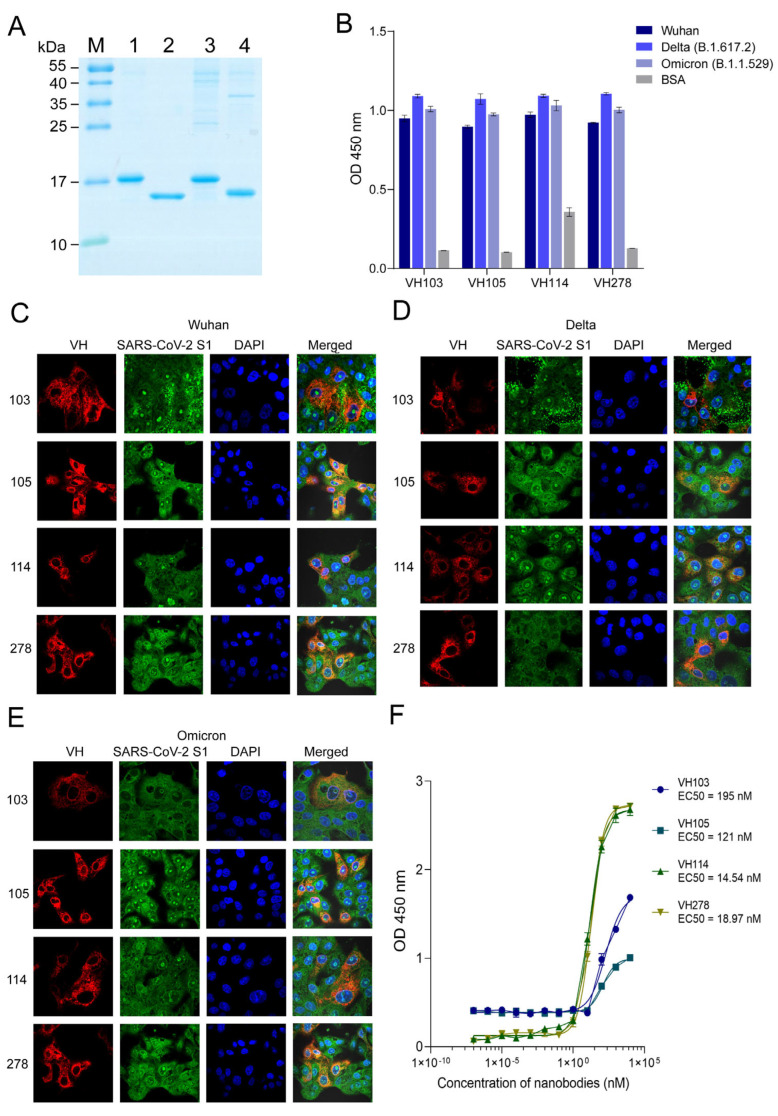
Purified 6× His-tagged nanobodies of *E. coli* clones 103, 105, 114, and 278 (VH103, VH105, VH114, and VH278, respectively) and their antigen binding: (**A**) Purified nanobodies from inclusion bodies of the *vh*-pET23b+ plasmid-transformed NiCo21 (DE3) *E. coli* clones 103, 105, 114, and 278 (lanes 1–4, respectively). M, protein standard marker. The numbers on the left are protein masses in kDa. (**B**) Indirect ELISA for testing the binding of 6× His-tagged VH103, VH105, VH114, and VH278 to recombinant Wuhan, Delta, and Omicron RBDs, using BSA as a control antigen. (**C**–**E**) The 6× His-tagged VH103, VH105, VH114, and VH278, respectively, bound to native S1 subunits of spike proteins of SARS-CoV-2 Wuhan wildtype and Delta and Omicron variants, as determined by confocal microscopy. Nanobodies stained red; native S1 subunits of the SARS-CoV-2 spike protein stained green; nuclei stained blue; co-localized VHs and S1 subunits in merged panels stained orange/yellow. (**F**) Half-maximal effective concentrations (EC50) of VH103, VH105, VH114, and VH278 against the recombinant S1 subunit.

**Figure 3 viruses-15-01252-f003:**
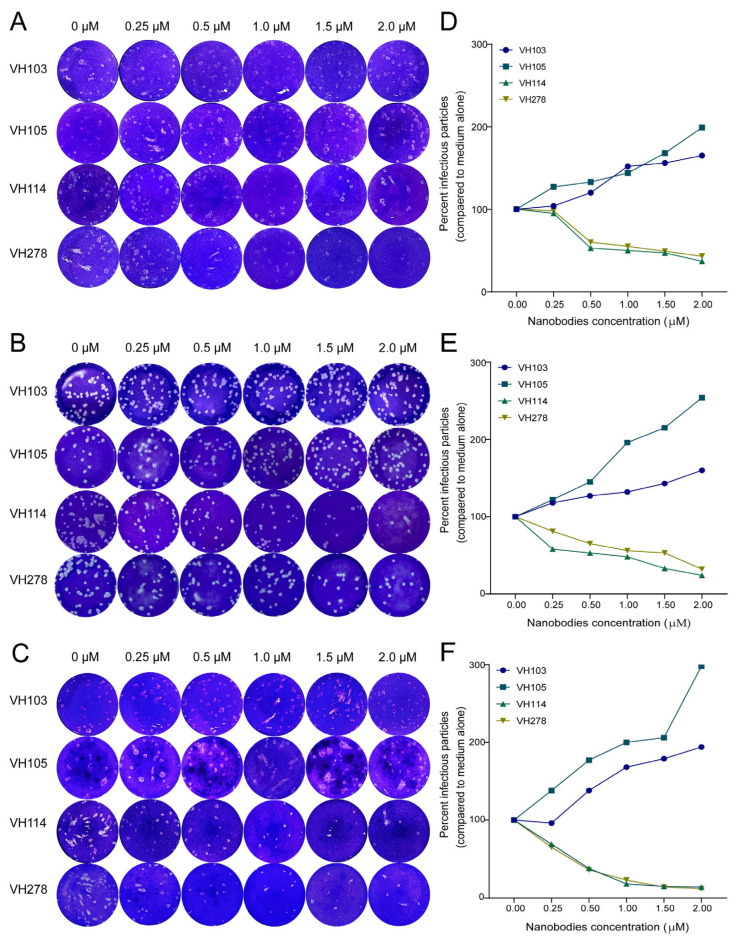
Nanobody-mediated neutralization/enhancement of SARS-CoV-2 infectivity: (**A**–**C**) Plaques of SARS-CoV-2 Wuhan, Delta, and Omicron, respectively, that were treated with different concentrations (0, 0.25, 0.5, 1.0, 1.5, and 2.0 μM) of nanobodies (VH103, VH105, VH114, and VH278) before being added to a Vero E6 cell monolayer and incubated for 5 days; data are from one of three independent and reproducible experiments. (**D**–**F**) Percentages of infectious particles (means ± standard deviation) of SARS-CoV-2 Wuhan, Delta, and Omicron BA1, respectively, after treatment with different concentrations of nanobodies, compared to the respective viruses in medium alone (0 μM of nanobodies); data are of three independent and reproducible experiments.

**Figure 4 viruses-15-01252-f004:**
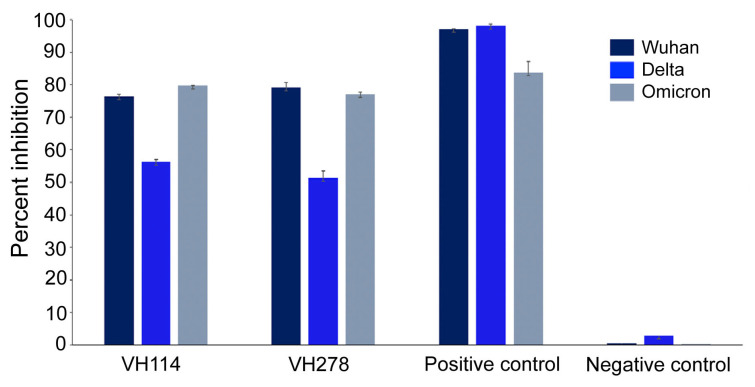
Interactions of SARS-CoV-2 RBDs with human ACE2 were inhibited by neutralizing VH114 and VH278, as tested by competitive ELISA. Mixtures of nanobodies and HRP-conjugated RBDs were added to immobilized human ACE2 in appropriate ELISA wells. Positive inhibition (human anti-SARS-CoV-2 RBD) and negative (background) inhibition (PBS-T) controls were also included in the assay. After incubation, the wells were washed, and a chromogenic substrate was added to each well for color development. The percentage inhibition of RBD binding to human ACE2 was calculated using the OD of the negative controls as no inhibition.

**Figure 5 viruses-15-01252-f005:**
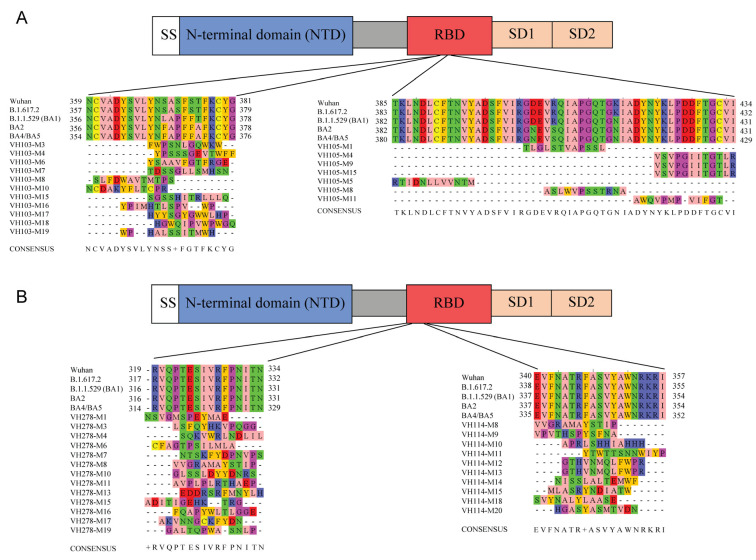
Amino acid sequences of the SARS-CoV-2 S1 subunit that were bound by nanobodies VH114, VH278, VH103, and VH105 (presumptive epitopes), as determined by phage mimotope identification using the Ph.D.^TM^-12 Phage display peptide library as a tool: (**A**) Domain organization of the SARS-CoV-2 S1 protein (adapted from [10]), and multiple alignment of the phage mimotopic sequences (M types) of enhancing VH103 (left panel) and VH105 (right panel) with the S1 RBD sequences. (**B**) Domain organization of the S1 protein of SARS-CoV-2. Mimotope peptide sequences of the phages that bound by neutralizing VH114 (right panel) and VH278 (left panel) were multiply aligned with the S1 RBD sequences to find the consensus sequences of the S1 RBD that matched the phage mimotopes (i.e., VH114 and VH278 presumptive epitopes). Other M types did not match the S1 RBD sequence, or the phages did not have inserted 12-mer peptide-coding DNAs (non-specific binding phages). SS, Signal sequence; NTD, N-terminal domain of S1 subunit; RBD, receptor-binding domain; SD1, subdomain 1; SD2, subdomain 2.

**Figure 6 viruses-15-01252-f006:**
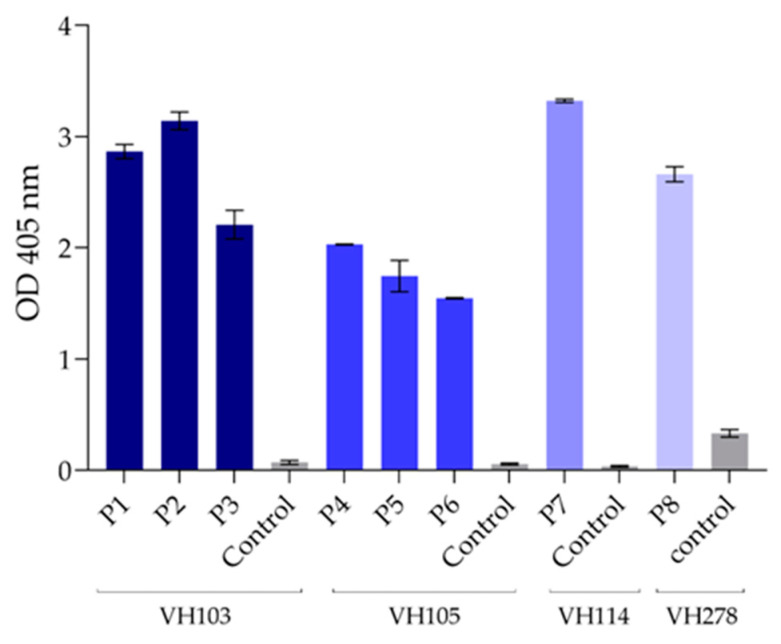
Peptide-binding ELISA to demonstrate binding of the nanobodies to the RBD peptides. The nanobodies VH103 bound to VH103-P1 (P1), VH103-P2 (P2), and VH103-P3 (P3); VH105 bound to VH105-P4 (P4), VH105-P5 (P5), and VH105-P6 (P6); VH114 bound to VH114-P7 (P7); and VH278 bound to VH278-P8 (P8), which verified that the consensus peptides contained epitopes of the respective nanobodies. Control (irrelevant) peptide was included in the assay as negative control.

**Figure 7 viruses-15-01252-f007:**
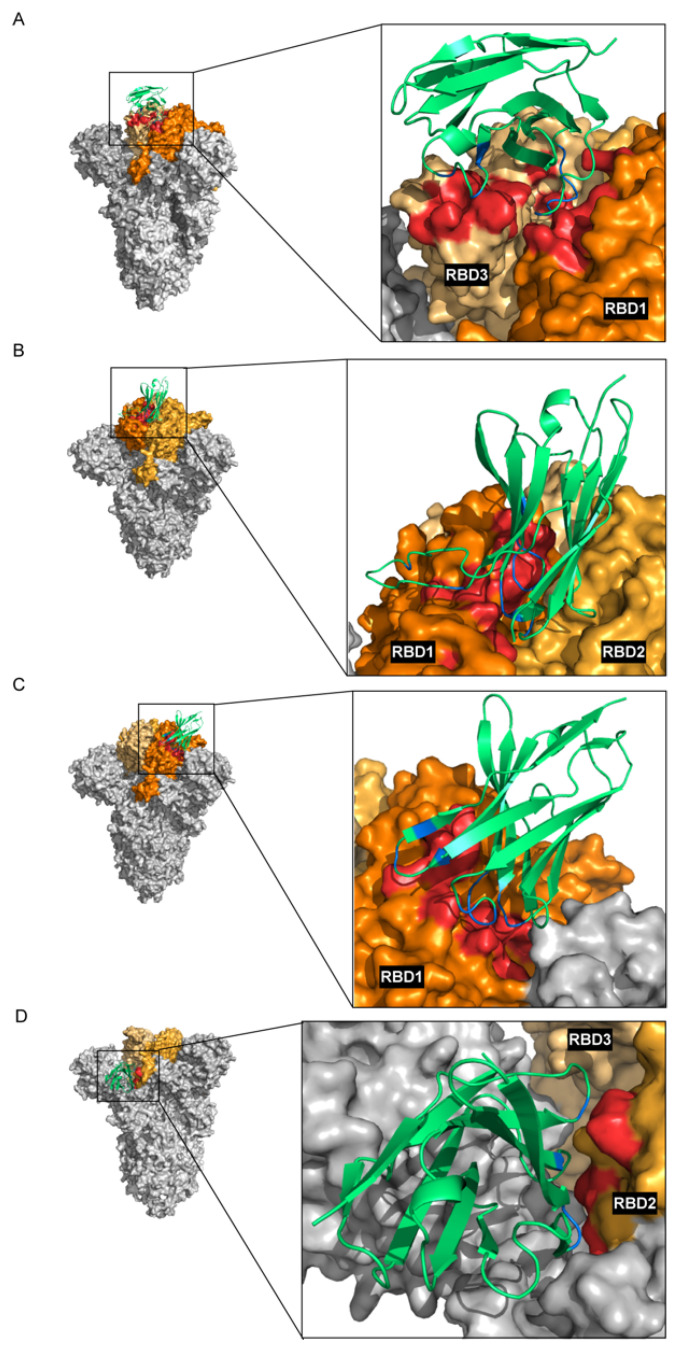
Computerized simulation to demonstrate interactions between SARS-CoV-2’s trimeric spike (S) protein and the nanobodies: (**A**–**D**) The contact interfaces between RBDs and VH103, VH105, VH114, and VH278, respectively. Monomers 1–3 of the trimeric S protein are shown in grey. RBDs are in orange shades, and the VHs are in lime green. The interacting residues between RBDs and VHs are shown in red and blue, respectively.

**Table 1 viruses-15-01252-t001:** Percentage homology of the framework sequences (FRs) of the nanobodies (VHs/V_H_Hs) from the 16 phagemid-transformed HB2151 *E. coli* clones to the closest human V FRs, and sequences and numbers of amino acids in their CDR3.

Sequence ID	Closest Human V Region	Identity (%)	Amino Acid Homology to Human FRs (%)			Sequence and Number of Amino Acids in CDR3	
			FR1	FR2	FR3	Sequence	Number of Residues
VH103	IGHV3-66*02 F	92.63	96.00	94.12	94.12	AGGAQDYSDYDDASLPTSMDY	21
VH105	IGHV3-7*05 F	98.26	100.00	100.00	100.00	ARVPETTVTTGPLPSYYYGMDV	22
VH114	IGHV3-66*02 F	89.12	96.00	88.24	88.24	ASPYQSINL	9
VH148	IGHV3-66*02 F	90.18	100.00	88.24	88.24	TRAVDYSIDY	10
VH156	IGHV4-39*07 F	81.79	84.00	82.35	82.35	ARVGADGSRFGGIDFDS	17
VH160	IGHV3-7*05 F	98.26	96.00	100.00	100.00	ARVPETTVTTGPLPYYYYGMDV	22
VH162	IGHV3-66*02 F	91.58	96.00	94.12	94.12	ATDRSGMWWRPA	12
VH184	IGHV3-7*05 F	98.96	100.00	100.00	100.00	ARVPETTVTTEPLPYYYYGMDV	22
VH187	IGHV3-66*02 F	90.18	88.00	94.12	94.12	ATSYDNDYALHPYNY	15
VH215	IGHV3-7*05 F	98.26	100.00	100.00	100.00	ARVPETTVTTGPLPYYYYGMDV	22
V_H_H219	IGHV3-66*02 F	83.86	80.00	52.94	52.94	AAGFSPTQPPYALRTSRYNY	20
VH228	IGHV3-66*02 F	88.42	96.00	82.35	82.35	AGIRRWDDGSWYTVERNVYNY	21
VH256	IGHV3-66*02 F	92.63	100.00	100.00	100.00	ARVPETTVTTGPLPYYYYGMDV	22
VH265	IGHV3-7*05 F	98.96	96.15	94.12	94.12	AAASDTIATMSAFGY	15
VH278	IGHV3-66*02 F	86.32	100.00	100.00	100.00	ARVPETTVTTGPLPYYYYGMDV	22
VH284	IGHV3-NL1*01 F	83.33	92.00	76.47	76.47	LRGGEGVY	8
VH285	IGHV3-7*05 F	98.61	96.00	70.59	70.59	ARSADYSIDY	10
						Max	22
						Min	8
						Mode	22

Asterisks followed by two numbers indicate the allele polymorphism; nt, nucleotides; FR, framework region of the immunoglobulin.

**Table 2 viruses-15-01252-t002:** Numbers (means ± standard deviation) of pfu in Vero E6 cells per well of the 24-well culture plates after adding mixtures of SARS-CoV-2 and various concentrations of nanobodies to the cells and incubating for 5 days. Data from three independent and reproducible experiments.

Nanobody	Nanobody Concentration (μM)	Pfu per Well			Percentage Neutralization		
		Wuhan	Delta	Omicron	Wuhan	Delta	Omicron
Enhancing VH103	0	25 ± 3.51	51 ± 1.00	26 ± 1.53	0.00	0.00	0.00
	0.25	25 ± 3.57	60 ± 5.13	28 ± 4.04	−2.70	−18.30	−7.79
	0.5	33 ± 3.21	65 ± 1.53	36 ± 2.00	−32.43	−26.80	−40.26
	1.0	40 ± 4.04	67 ± 3.06	43 ± 6.56	−63.51	−32.03	−67.53
	1.5	44 ± 2.12	74 ± 5.13	45 ± 4.51	−76.35	−44.44	−74.03
	2.0	51 ± 0.71	82 ± 3.51	52 ± 2.08	−104.73	−60.13	−103.90
Enhancing VH105	0	22 ± 4.04	30 ± 2.52	25 ± 3.61	0.00	0.00	0.00
	0.25	26 ± 0.58	36 ± 4.73	32 ± 1.53	−21.54	−22.47	−26.67
	0.5	28 ± 1.53	43 ± 5.00	35 ± 4.62	−27.69	−44.94	−41.33
	1.0	28 ± 2.65	58 ± 12.49	39 ± 3.79	−29.23	−95.51	−57.33
	1.5	32 ± 2.08	60 ± 2.08	45 ± 4.73	−49.23	−103.37	−81.33
	2.0	47 ± 5.03	75 ± 2.08	50 ± 1.53	−115.38	−153.93	−98.67
Neutralizing VH114	0	49 ± 8.33	56 ± 9.64	40 ± 4.04	0.00	0.00	0.00
	0.25	34 ± 5.29	32 ± 5.03	38 ± 4.16	31.08	42.26	5.04
	0.5	19 ± 4.51	30 ± 2.52	21 ± 9.17	62.16	47.02	47.06
	1.0	9 ± 3.06	27 ± 2.31	17 ± 7.77	82.43	52.38	56.30
	1.5	7 ± 2.08	18 ± 3.79	15 ± 2.52	85.14	67.26	61.34
	2.0	6 ± 1.00	13 ± 1.15	9 ± 4.04	87.84	76.19	78.15
Neutralizing VH278	0	55 ± 3.21	67 ± 6.43	34 ± 3.51	0.00	0.00	0.00
	0.25	36 ± 4.24	54 ± 3.51	30 ± 6.66	56.63	19.50	11.65
	0.5	20 ± 1.53	43 ± 2.65	19 ± 0.58	64.46	35.50	45.63
	1.0	13 ± 2.12	37 ± 1.53	14 ± 1.53	84.94	44.00	60.19
	1.5	8 ± 0.58	35 ± 1.00	9 ± 1.53	86.14	47.50	74.76
	2.0	7 ± 1.53	21 ± 4.36	7 ± 2.08	87.95	68.50	80.58

**Table 3 viruses-15-01252-t003:** Residues and regions of RBDs (monomers 1, 2, and 3) of the trimeric SARS-CoV-2 spike (S) protein predicted to form contact interfaces with the enhancing VH103 and VH105 and neutralizing VH114 and VH278.

**Trimeric S Protein**		**VH103**		**Interactive Bond**
**Residue**	**RBD Monomer Number**	**Residue**	**Domain**	
Arg403 *	1	Asp104	CDR3	Hydrogen bond; electrostatic
Tyr489 *	1	Ala108	CDR3	Hydrophobic
Arg493 *	1	Asp106	CDR3	Hydrogen bond; electrostatic
Leu371	3	Ala108	CDR3	Hydrophobic
Pro373	3	Tyr105	CDR3	Hydrophobic
Trp436	3	Asp104	CDR3	Hydrogen bond
Trp436	3	Tyr105	CDR3	Hydrophobic
Lys440 *	3	Tyr102	CDR3	Hydrogen bond
**Trimeric S Protein**		**VH105**		**Interactive Bond**
**Residue**	**RBD Monomer Number**	**Residue**	**Domain**	
Arg403	1	Glu52	CDR2	Hydrogen bond; electrostatic
Arg403	1	Asp54	CDR2	Hydrogen bond; electrostatic
Gly447 *	1	Glu57	CDR2	Hydrogen bond
Leu452 *	1	Pro110	CDR3	Hydrophobic
Phe456 *	1	His31	CDR1	Hydrophobic
Cys488 *	1	Tyr32	CDR1	Hydrogen bond
Tyr489 *	1	Thr28	CDR1	Hydrogen bond
Tyr489 *	1	His31	CDR1	Hydrophobic
Arg493 *	1	His31	CDR1	Hydrogen bond
Arg493 *	1	Gln53	CDR2	Hydrogen bond
Arg493 *	1	Glu101	CDR3	Electrostatic
Ser494 *	1	Gln53	CDR2	Hydrogen bond
Ser494 *	1	Glu101	CDR3	Hydrogen bond
Ser496 *	1	Glu52	CDR2	Hydrogen bond
Ser496 *	1	Ser56	CDR2	Hydrogen bond
Ser496 *	1	Glu57	CDR2	Hydrogen bond
Arg498 *	1	Glu57	CDR2	Hydrogen bond; electrostatic
Arg498 *	1	Lys58	CDR2	Hydrogen bond
Tyr501 *	1	Ser56	CDR2	Hydrogen bond
His505 *	1	Asp54	CDR2	Hydrogen bond
Leu371	2	His31	CDR1	Hydrophobic
**Trimeric S Protein**		**VH114**		**Interactive Bond**
**Residue**	**RBD Monomer Number**	**Residue**	**Domain**	
Glu340	1	Thr28	CDR1	Hydrogen bond
Glu340	1	Ser31	CDR1	Hydrogen bond
Glu340	1	Thr32	CDR1	Hydrogen bond
Asn343	1	Ser101	CDR3	Hydrogen bond
Ala344	1	Pro98	CDR3	Hydrophobic
Thr345	1	Asn103	CDR3	Hydrogen bond
Arg346	1	Tyr33	CDR1	Hydrophobic
Asn354	1	Ser52	CDR2	Hydrogen bond
Arg355	1	Asp54	CDR2	Hydrogen bond
Lys356	1	Ser30	CDR1	Hydrogen bond
Lys356	1	Pro53	CDR2	Hydrophobic
Arg466 *	1	Asp54	CDR2	Hydrogen bond; electrostatic
**SARS-CoV-2 RBD**		**VH278**		**Interactive Bond**
**Residue**	**RBD Monomer Number**	**Residue**	**Domain**	
Arg319	2	Gly54	CDR2	Hydrogen bond
Gln321	2	Trp33	CDR1	Hydrogen bond
Gln321	2	Ser53	CDR2	Hydrogen bond
Glu324	2	Gly99	CDR3	Hydrogen bond
Lys537	2	Trp33	CDR1	Hydrophobic

* Amino acid that binds tightly to human ACE2 [37].

## Data Availability

All datasets presented in this study are included in the article and Appendix A.

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
