# Peer review of "Neutralizing and Enhancing Epitopes of the SARS-CoV-2 Receptor-Binding Domain (RBD) Identified by Nanobodies"

_viruses, 2023, doi:10.3390/v15061252_

Round 1

Reviewer 1 Report

  • Major comments: 

In this study, Kanasap Kaewchim et al. screened a humanized-camel VH/VHH-phage display library using recombinant Wuhan RBD as bait and found sixteen phage infected-E. Coli clones produced nanobodies with similar frameworks to human antibodies. Of them, some nanobodies can neutralize the SARS-CoV-2 virus while others enhance the virus infectivity in Vero E6 cells. They further identified the epitopes of these nanobodies.

The study is helpful for a better understanding of the neutralizing and enhancing epitopes of SARS-CoV-2 RBD.

  • General concept comments

Here are some major considerations/suggestions for the study:

1.      The mechanisms of the neutralizing effects of Nanobodies clones 114 and 278 should be determined. For example, do nanobodies clones 114 and 278 interfere with the binding of SARS-CoV-2 RBD to the ACE2 receptor?

2.      As ADE is greatly concerned, the mechanisms of ADE of nanobodies clones 103 and 105 should be identified.

3.      Although the study identified the epitopes in the RBD for the four nanobodies, additional experiments are needed to confirm these epitopes if the structure data is not available. For example, the RBD lacking these epitopes should be purified and tested for whether these RBDs can still bind to the four nanobodies, respectively.

4.      In the section of 2.5. Production of the RBD-Bound Nanobodies in Large Scale, it seems like these nanobodies are not soluble. Did you try to purify the nanobodies from the soluble part of bacterial rather than inclusion bodies? How about the yield of the nanobodies if purified from inclusion bodies?

5.      In the section of 2.6. Indirect enzyme-linked immunosorbent assay, it has been shown that the Optical densities were measured either at 405 nm or 450 nm. Could you please specify why? Why don't you use the same Elisa procedure?

  • Specific comments:

1)       In the C, D, and E panels of Figure 2, it would be better to label the strain name of each SARS-CoV-2.

2)       In the D, E, and F panels of Figure 3, it would be better to do the statistical analysis.

Fine.

Author Response

Response to comments and suggestions of Reviewer 1

  • Major comments: 

In this study, Kanasap Kaewchim et al. screened a humanized-camel VH/VHH-phage display library using recombinant Wuhan RBD as bait and found sixteen phage infected-E. coli clones produced nanobodies with similar frameworks to human antibodies. Of them, some nanobodies can neutralize the SARS-CoV-2 virus while others enhance the virus infectivity in Vero E6 cells. They further identified the epitopes of these nanobodies. 

The study is helpful for a better understanding of the neutralizing and enhancing epitopes of SARS-CoV-2 RBD.

  • General concept comments: Here are some major considerations/suggestions for the study:

  1. The mechanisms of the neutralizing effects of Nanobodies clones 114 and 278 should be determined. For example, do nanobodies clones 114 and 278 interfere with the binding of SARS-CoV-2 RBD to the ACE2 receptor?

Response:

We have performed a competitive ELISA (section 2.11 of the revised manuscript R1).  The results in section 3.4 showed that the neutralizing nanobodies of both clones (clones 114 and 278) inhibited binding of the RBDs to human ACE2 as shown in Figure 4 of the revised manuscript. Thank you very much.

  1. As ADE is greatly concerned, the mechanisms of ADE of nanobodies clones 103 and 105 should be identified.

Response:

      Your suggestion is very well taken. We will identify the ADE mechanisms of the VH103 and VH105 and will publish the information later. For the time being, we have added discussion on the possible ADE mechanisms of VH103 and VH105 in the last paragraph of the Discussion section of the revised manuscript. Thank you very much.

  1. Although the study identified the epitopes in the RBD for the four nanobodies, additional experiments are needed to confirm these epitopes if the structure data is not available. For example, the RBD lacking these epitopes should be purified and tested for whether these RBDs can still bind to the four nanobodies, respectively.

Response:

     In the revised manuscript, we have provided computerized simulation to demonstrate interaction of the four nanobodies to the RBD of trimeric spike (S) protein of SARS-CoV-2 (Figure 7 and Table 2 of the revised manuscript). We apologize that we cannot perform the experiments using RBD that lacks the target epitopes this time. But your point is very well taken with many thanks.

  1. In the section of 2.5. Production of the RBD-Bound Nanobodies in Large Scale, it seems like these nanobodies are not soluble. Did you try to purify the nanobodies from the soluble part of bacteria rather than inclusion bodies? How about the yield of the nanobodies if purified from inclusion bodies?

Response:

The nanobodies were expressed in soluble form in minute (negligible) amounts compared to those in the inclusion bodies (IB) as shown in the below Figure (clone 103 is an example). Therefore, we did not purify them from the soluble parts, as rather large amounts were required for all experiments (we used one batch of nanobodies).

The IB from one liter of E. coli culture was approximately 20 grams; after the IB was purified and refolded, about 500 mg of purified nanobodies were obtained. However, it varies depending on the transformed E. coli clones. Thank you very much.

  1. In the section of 2.6. Indirect enzyme-linked immunosorbent assay, it has been shown that the Optical densities were measured either at 405 nm or 450 nm. Could you please specify why? Why don't you use the same Elisa procedure?

Response:

       We used different substrates in the ELISAs because the commercially available secondary antibodies (anti-primary antibody isotypes raised in different animal species) were conjugated with different enzymes; therefore, different substrates had to be used. The optical densities were determined at different wavelengths based on the kind of substate, as instructed by the manufacturers. Thank you very much.

  • Specific comments:
  • In the C, D, and E panels of Figure 2, it would be better to label the strain name of each SARS-CoV-2.

Response:

The virus strain/variants are placed in Figures 2C, 2D and 2E as per your suggestion. Thank you very much.

  • In the D, E, and F panels of Figure 3, it would be better to do the statistical analysis.

Response:

We have provided the statistical analysis of Figures 3D, 3E and 3F in Supplementary Table 1. Thank you very much.  

On behalf of all authors, I would like to thank you very much for your time spent on reading our manuscript and your valuable comments and suggestions.

  Yours Sincerely,

Wanpen Chaicumpa, D.V.M. (Hons.), Ph. D.

Reviewer 2 Report

In the paper “Neutralizing and Enhancing Epitopes of SARS-CoV-2 Receptor-Binding Domain (RBD) Identified by Nanobodies”, the authors Kaewchim et al. describe the development of nanobodies capable of binding SARS-CoV-2 RBD protein, with either neutralizing or enhancing effect on the virus. The study uses what the authors call humanized-camel VH/VHH phage display library to isolate nanobodies specific to Wuhan RBD protein of SARS-CoV-2. Four such nanobodies were selected, which also bind the Delta and Omicron variants. Two of these clones were found to be neutralizing in an in vitro assay and two were found to enhance the virus infectivity. Some epitope mapping was conducted to define each nanobody epitope on the RBD.

General comments:

1.       Was the library produced from naïve camels? If so, it is quite remarkable that the authors were able to select several specific clones after one round of panning. Can the authors suggest some explanation for this fact?

2.       The authors claim throughout the text that this is a humanized library, however, no humanization steps were carried here. The authors merely selected for nanobodies with similarity to human sequences, at the 5’ and 3’ ends. This cannot be regarded as humanization. Perhaps “human-like” is a more appropriate definition.

3.       No binding characteristics are shown, to asses how well these nanobodies bind the RBD, and if there is any difference in their affinity towards the different variants’ RBDs. The same ELISA as shown in figure 2B could be done, using serial dilutions of each nanobody, to get a better understanding of their binding affinity.

4.       Numerous, fully-human antibodies to SARS-CoV-2 were isolated and generated in the last few years. The authors should emphasize what is the advantage of their nanobodies. Only at the end of discussion the authors mention that “…small sized antibody that usually has deep and high penetrating activity compared to intact four-chain antibody molecules” (line 683). If this is one major advantage of nanobodies, it should be stated earlier.

5.       This paper displays a very initial and basic evaluation of the selected nanobodies, yet throughout the discussion they strongly determine that these “should be tested and developed further for use as a safe passive immunotherapeutic agent” (line 665), “this epitope should be included in SARS-CoV-2 subunit vaccine” (685-686), and so on. Such conclusions should be moderated. Also, a more comprehensive comparison to other developed antibodies would be of benefit to the text.

6.       Neutralization should be displayed as NT50 values or at least as neutralization percent, so a real evaluation of potency could be obtained and comparison to other reported antibodies will be possible. Plaques numbers are not informative enough.

Specific comments:

1.       The study uses only gene fragments that were amplified by human Ig primers. What was the size of the obtained library? What percentage of the camel VHs were amplified?

2.       Lines 403-404: “The length of the CDR3 of the humanized-nanobodies ranged between 8–22 amino acid residues, with the mode at 22 residues (Table 1).” This data is not presented in Table 1.

3.       Lines 404-405: “The FR2 of clone 219 revealed characteristic amino acid tetrad of VHH”. What does that mean? Either explain or discard.

4.       Figure 1: the legend states that “Asterisks indicate the E. coli clones that their expressed nanobodies gave positive indirect ELISA results (OD 405 nm to RBD was > two times higher than the OD 405 nm to BSA).” However, clones 148, 162 and 219 do not display two times higher binding compared to BSA.

5.       Figure 3: Plaques in A and C seem different from those in B. Can the authors explain this? Are these at all real plaques? Also, to the naked eye, VH103 does not display real difference in A and B.

6.       Table 2: While data of one representing experiments can be displayed in a figure, the table should present the average results of all three experiments.

Why is there a difference in the amount of plaques observed with no nanobody (0 µM) between the enhancing (22-25 plaques for Wuhan) and neutralizing nanobodies (49-55 plaques)? In fact, the enhancing nanobodies display in the highest concentration tested, the same amount of plaques as displayed in the 0 µM samples of the neutralizing nanobodies. Could the enhancing effect that the authors see is merely a side effect of the assay?

7.       Lines 634-636: “the soluble E-tagged-nanobodies in the lysates of phage transformed HB2151 E. coli clones 114 and 278 (VH114 and VH278) strongly neutralized infectivity of both SARS-CoV-2 strains”. This is not shown. The neutralization assay that is shown does not display any values that allow understanding if this is indeed a strong neutralization. Such statements should be refined.

8.       It is not clear if the phage mimotope are linear epitopes or considered conformational.

9.       Lines 699-701: “The findings indicate that the epitope of enhancing VH105 should be a conformational epitope that is formed by amino acids of the three RBD regions that are spatially juxtaposed upon the protein folding.” Usually, when an epitope is conformational, linear parts of it will not be recognized by the antibody. The VH105 epitope may be conformational but the binding observed may not be specific. This conclusion therefore should be more careful.   

Author Response

Response to Reviewer’s 2 comments and suggestions

In the paper “Neutralizing and Enhancing Epitopes of SARS-CoV-2 Receptor-Binding Domain (RBD) Identified by Nanobodies”, the authors Kaewchim et al. describe the development of nanobodies capable of binding SARS-CoV-2 RBD protein, with either neutralizing or enhancing effect on the virus. The study uses what the authors call humanized-camel VH/VHH phage display library to isolate nanobodies specific to Wuhan RBD protein of SARS-CoV-2. Four such nanobodies were selected, which also bind the Delta and Omicron variants. Two of these clones were found to be neutralizing in an in vitro assay and two were found to enhance the virus infectivity. Some epitope mapping was conducted to define each nanobody epitope on the RBD.

General comments:

  1. Was the library produced from naïve camels? If so, it is quite remarkable that the authors were able to select several specific clones after one round of panning. Can the authors suggest some explanation for this fact?

Response:                                                                                                                                                                                                 Yes, the library was constructed by using cDNA prepared from total RNA extracted from peripheral blood mononuclear cells of naïve C. dromedarius. Oligonucleotide primers specific to all families of human VH and JH genes (forward and reverse primers respectively) were used to amplify the cDNA templates. The camel VH/VHH gene segments could be amplified by nine combinations of the forward and reverse human immunoglobulin-specific primers, i.e., VH1a + JH1245, VH1a + JH3, VH1c + JH1245, VH1d + JH1245, VH1d + JH3, VH3a + JH1245, VH3a + JH3, VH3b + JH1245, and VH3b + JH3. The vhs/vhhs amplicons were pooled and used in the library construction. A total of ∼4 × 1011 VH/VHH-displaying mature phage particles were obtained after co-infecting the phagemid transfected TG1 E. coli with M13K07 helper phages in the phage-rescuing process. By using restriction fragment length polymorphism (RFLP) analysis, more than 80% of the phages revealed different fragment patterns. Therefore, the antibody diversity of the phages in the library should be > 3.2 × 1011 (Thanongsaksrikul J et al. J Biol Chem 2010; 285: 9657-9666). We think RBD is a foreign immunodominant viral protein, therefore, many RBD binding phage clones were obtained.

  1. The authors claim throughout the text that this is a humanized library, however, no humanization steps were carried here. The authors merely selected nanobodies with similarity to human sequences, at the 5’ and 3’ ends. This cannot be regarded as humanization. Perhaps “human-like” is a more appropriate definition.

Response:

      Thank you very much for your suggestion. We have changed the word “humanized” to “human-like” or deleted the word “humanized” as per your advice.

  1. No binding characteristics are shown to assess how well these nanobodies bind the RBD, and if there is any difference in their affinity towards the different variants’ RBDs. The same ELISA as shown in figure 2B could be done, using serial dilutions of each nanobody, to get a better understanding of their binding affinity.

Response:

        EC50 of the nanobodies are provided in the revised manuscript (R1). Thank you very much.

  1. Numerous, fully human antibodies to SARS-CoV-2 were isolated and generated in the last few years. The authors should emphasize what is the advantage of their nanobodies. Only at the end of discussion the authors mention that “…small sized antibody that usually has deep and high penetrating activity compared to intact four-chain antibody molecules” (line 683). If this is one major advantage of nanobodies, it should be stated earlier.

Response:

We have moved the statements on the advantages of nanobodies to the “Introduction” section and added other characteristics of the nanobodies as per your advice. Thank you very much.

  1. This paper displays a very initial and basic evaluation of the selected nanobodies, yet throughout the discussion they strongly determine that these “should be tested and developed further for use as a safe passive immunotherapeutic agent” (line 665), “this epitope should be included in SARS-CoV-2 subunit vaccine” (685-686), and so on. Such conclusions should be moderated. Also, a more comprehensive comparison to other developed antibodies would be of benefit to the text.

Response:

Thank you very much. We rephrased the sentences.

  1. Neutralization should be displayed as NT50 values or at least as neutralization percent, so a real evaluation of potency could be obtained and comparison to other reported antibodies will be possible. Plaques numbers are not informative enough.

Response:

The neutralization percentages of the nanobodies are added to Table 2 of the revised manuscript. Thank you very much.

Specific comments:

  1. The study uses only gene fragments that were amplified by human Ig primers. What was the size of the obtained library? What percentage of the camel VHs were amplified?

Response:

       Please see the response to your comment 1. Thank you very much.

  1. Lines 403-404: “The length of the CDR3 of the humanized-nanobodies ranged between 8–22 amino acid residues, with the mode at 22 residues (Table 1).” This data is not presented in Table 1.

Response:

We have added the amino acid sequences and numbers of amino acids of CDR3 of the nanobodies in Table 1. Thank you very much.

  1. Lines 404-405: “The FR2 of clone 219 revealed characteristic amino acid tetrad of VHH”. What does that mean? Either explain or discard.

Response:

Thank you very much. We have explained the characteristic tetrads of VHH in the revised manuscript.

  1. Figure 1: the legend states that “Asterisks indicate the coli clones that their expressed nanobodies gave positive indirect ELISA results (OD 405 nm to RBD was > two times higher than the OD 405 nm to BSA).” However, clones 148, 162 and 219 do not display two times higher binding compared to BSA.

Response:

We rephrased the sentence. Thank you very much.

  1. Figure 3: Plaques in A and C seem different from those in B. Can the authors explain this? Are these at all real plaques? Also, to the naked eye, VH103 does not display real difference in A and B.

 Response:

        Different SARS-CoV-2 variants have different infectivity and virulence. The Delta is highly infectious. We and other colleagues noticed that the plaques of the different SARS-CoV-2 strains are different. It was very difficult to take clear pictures of the plaques.

  1. Table 2: While data of one representing experiment can be displayed in a figure, the table should present the average results of all three experiments.

Response:

        We apologize for the errors. Figures 3A-3C were from one experiment as a representative of the many replicative experiments. Data of Figures 3D-3E and Table 2 were from three independent and replicative experiments.

Why is there a difference in the amount of plaques observed with no nanobody (0 µM) between the enhancing (22-25 plaques for Wuhan) and neutralizing nanobodies (49-55 plaques)? In fact, the enhancing nanobodies display in the highest concentration tested, the same amount of plaques as displayed in the 0 µM samples of the neutralizing nanobodies. Could the enhancing effect that the authors see is merely a side effect of the assay?

Response:

        Different aliquots may contain different numbers of virus particles.

        The experiments were repeated many times and the overall results were reproducible. We are confident that VH103 and VH105 had an enhancing effect in a dose-dependent manner. Thank you very much.

  1. Lines 634-636: “the soluble E-tagged-nanobodies in the lysates of phage transformed HB2151 coli clones 114 and 278 (VH114 and VH278) strongly neutralized infectivity of both SARS-CoV-2 strains”. This is not shown. The neutralization assay that is shown does not display any values that allow understanding if this is indeed a strong neutralization. Such statements should be refined.

Response:

Thank you very much. We have deleted “strongly” from the sentence.

  1. It is not clear if the phage mimotope are linear epitopes or considered conformational.

Response:

Phage mimotope are linear sequences.

  1. Lines 699-701: “The findings indicate that the epitope of enhancing VH105 should be a conformational epitope that is formed by amino acids of the three RBD regions that are spatially juxtaposed upon the protein folding.” Usually, when an epitope is conformational, linear parts of it will not be recognized by the antibody. The VH105 epitope may be conformational but the binding observed may not be specific. This conclusion therefore should be more careful.

Response:   

Thank you very much for your comment.

 May I have the following thought, although I cannot prove it for the time being.

       I agree with you that many (conventional) antibodies to conformational epitopes do not bind their target proteins in the assays that require whole or partial denaturation of the proteins (linear parts of the protein) such as Western blot assay (Forsström, B., Axnäs, B.B., Rockberg, J., Danielsson, H., Bohlin, A., Uhlen M. Dissecting antibodies with regards to linear and conformational epitopes. PLoS One 2015; 10(3): e0121673. doi: 10.1371/journal.pone.0121673). Nevertheless, the VH105 bound to the RBD consensus peptides, i.e., biotin-labelled- VH105-P4, VH105-P5, and VH105-P6 in the peptide-binding ELISA. In the peptide-binding ELISA, the target was linked to streptavidin-hACE2 immobilized on the well surface via biotin, making it accessible easily by the antibodies. By computerized simulation of intermolecular docking, VH105 was predicted to use all three CDRs to interact with RBD of the SARS-CoV-2 spike protein (Figure 7 and Table 2). Camelid VHs typically have long CDR3s (CDR3 of VH105 contains 22 residues; longer than conventional VH CDR3). The camelid VH usually lack the salt bridge between Arg94 and Asp101 but may have additional Cys residues in both the CDR1 and CDR3 for forming a second internal disulfide bridge to adopt a secondary structure that is different from the canonical structure of the conventional hypervariable region. Besides, the camelid paratope tends to be larger than that of a VH-VL paratope, which enables the former to fit well into the concave-shaped cavity of the antigen surface. The differences in structural features of the paratopes may/might contribute to the different capacity in target binding.

On behalf of all authors, I would like to thank you very much for your time spent on reading our manuscript and your valuable comments and suggestions.

  Yours Sincerely,

Wanpen Chaicumpa, D.V.M. (Hons.), Ph. D.

Round 2

Reviewer 1 Report

I think that the manuscript has been improved, and the authors have addressed most of my concerns.

Fine.

Reviewer 2 Report

I have no further comments.